# Elucidating the structure-stability relationship of Cu single-atom catalysts using *operando* surface-enhanced infrared absorption spectroscopy

Li Zhang[1,2], Xiaoju Yang[1,2], Qing Yuan[1,2], Zhiming Wei[3], Jie Ding [ORCID][3], Tianshu Chu[4], Chao Rong [ORCID][4], Qiao Zhang[3], Zhenkun Ye[1,2], Fu-Zhen Xuan[4], Yueming Zhai [ORCID][3], Bowei Zhang [ORCID][4] ✉ & Xuan Yang [ORCID][1,2] ✉

Understanding the structure-stability relationship of catalysts is imperative for the development of high-performance electrocatalytic devices. Herein, we utilize *operando* attenuated total reflectance surface-enhanced infrared absorption spectroscopy (ATR-SEIRAS) to quantitatively monitor the evolution of Cu single-atom catalysts (SACs) during the electrochemical reduction of $CO_2$ ($CO_2$RR). Cu SACs are converted into 2-nm Cu nanoparticles through a reconstruction process during $CO_2$RR. The evolution rate of Cu SACs is highly dependent on the substrates of the catalysts due to the coordination difference. Density functional theory calculations demonstrate that the stability of Cu SACs is highly dependent on their formation energy, which can be manipulated by controlling the affinity between Cu sites and substrates. This work highlights the use of *operando* ATR-SEIRAS to achieve mechanistic understanding of structure-stability relationship for long-term applications.

With the development of human society, excessive amount of greenhouse gas $CO_2$ has been released because of large-scale utilization of fossil fuels, which causes severe issues including ocean acidification and climatic change[1,2]. Conversion of $CO_2$ into value-added chemicals and fuels enabled by electrochemical reduction of $CO_2$ ($CO_2$RR) has drawn great attention for reducing $CO_2$ emissions and realizing carbon recycling[3,4]. In the past few decades, much research effort has been dedicated into developing highly active catalysts for $CO_2$RR. Because of the theoretical maximum utilization efficiency, single-atom catalysts (SACs) have shown great potential for $CO_2$RR with superior activity and selectivity[5–7]. The unique coordination of the active sites (M-N$_4$) decreases the activation barrier for the formation of intermediate *COOH and the C-C coupling is greatly suppressed due to the low

coverage of *CO intermediates on highly dispersed metal sites, thus leading to superior selectivity of $C_1$ products[8,9]. Despite the extensively exploring of structure-activity relationships, the corresponding structure-stability relationships of SACs during $CO_2$RR are still lacking. The stability of SACs involves the compositional, structural, and morphological integrity of the material, which is closely related to their $CO_2$RR performance. Therefore, it is crucial to understand and control the stability of SACs, and thus to optimize the overall $CO_2$RR performance.

Recent studies have shown that $C_{2+}$ products (e.g., $C_2H_4$, $C_2H_6$, and $C_2H_5OH$) are generated on Cu SACs during the $CO_2$RR[10–12], which seems to be conflict to previous reports that C-C coupling for the formation of $C_{2+}$ products is prohibited on the SACs[13–16]. Density

[1]Key Laboratory of Material Chemistry for Energy Conversion and Storage, Huazhong University of Science and Technology, Wuhan 430074, China. [2]Hubei Key Laboratory of Bioinorganic Chemistry and Materia Medica, School of Chemistry and Chemical Engineering, Huazhong University of Science and Technology, Wuhan 430074, China. [3]The Institute for Advanced Studies, Wuhan University, Wuhan 430072, China. [4]Shanghai Key Laboratory of Intelligent Sensing and Detection Technology, Key Laboratory of Pressure Systems and Safety of Ministry of Education, School of Mechanical and Power Engineering, East China University of Science and Technology, Shanghai 200237, China. ✉e-mail: boweiz@ecust.edu.cn; xuanyang@hust.edu.cn

functional theory (DFT) calculations demonstrate that the Cu single site coordinated with four pyrrole-N atoms is the main active site for the production of acetone and reduces the reaction free energies required for $CO_2$ activation[14]. In contrast, *operando* X-ray absorption spectroscopy (XAS) studies reveal that isolated sites transiently convert into metallic Cu nanoparticles during the electrolysis, which are likely the active phase for the formation of $C_{2+}$ products[17]. It is well known that standard XAS techniques are not intrinsically surface sensitive, which causes difficulties to quantitatively monitor the surface reconstruction of the metal sites in SACs[18]. To ascertain the active sites and better understand the structure-activity relationships, it is of great importance to quantitatively correlate the $CO_2$RR performance and structure of catalysts under the reaction conditions.

Herein, we carry out *operando* attenuated total reflectance surface-enhanced infrared absorption spectroscopy (ATR-SEIRAS) to quantitatively monitor the reconstruction processes of several Cu SACs during the $CO_2$RR (Fig. 1a). Specifically, the Cu single sites of the Cu SACs ($Cu/C_3N_4$, CuPc, Cu-NC, and Cu-SNC) are anchored on carbon nitride, phthalocyanine, N doped carbon matrix, and N and S modified carbon matrix, respectively. The conversion rate of Cu single sites to nanoparticles is found to be heavily dependent on the applied potentials, with drastically increased rates at low potentials. The formation rate of metallic Cu sites at −1.2 V vs RHE (all potentials are referenced to reversible hydrogen electrode, RHE) is around two orders of magnitude higher than that at −0.6 V. The stability of Cu SACs correlates well with the affinity between the Cu sites and catalyst substrates, specifically, strong interactions lead to high stability. Among the four different types of Cu SACs, the $Cu/C_3N_4$ catalysts are the least stable, with significant reconstruction after the $CO_2$RR at −1.2 V for 20 min. The evolution rate of the $Cu/C_3N_4$ catalysts is determined to be $1.35 \times 10^{-3}$ min$^{-1}$ at −1.2 V based on the SEIRAS results. Meanwhile, reconstructions are observed in the CuPc and Cu-NC catalysts after electrolysis at −1.2 V for 60 and 165 min, with moderate evolution rates of $6.87 \times 10^{-4}$ min$^{-1}$ and $2.97 \times 10^{-4}$ min$^{-1}$, respectively. Cu-SNC catalysts stay stable during the $CO_2$RR at −1.2 V for 225 min and quantitative SEIRAS results show an evolution rate of $1.18 \times 10^{-4}$ min$^{-1}$ for the Cu-SNC catalysts at −1.2 V. Reactivity results demonstrate that the reconstructed Cu nanoparticles during the $CO_2$RR are critical to the formation of $C_{2+}$ products. No $C_{2+}$ products but CO is produced during the $CO_2$RR before Cu SACs evolves into Cu nanoparticles. The combined experimental and DFT results reveal that the stability of Cu SACs is dependent on the formation energy of Cu-X sites (X refers to C, N, and S etc.), which can be manipulated by controlling the coordination of Cu sites and the affinity between substrates and Cu sites. Therefore, our work demonstrates the successful application of *operando* ATR-SEIRAS in quantitatively monitoring the evolution of Cu SACs during the $CO_2$RR, which provides a novel approach to understanding the structure-stability relations of Cu SACs and the feasibility of improving the stability of Cu SACs by manipulating the coordination environments of the catalysts.

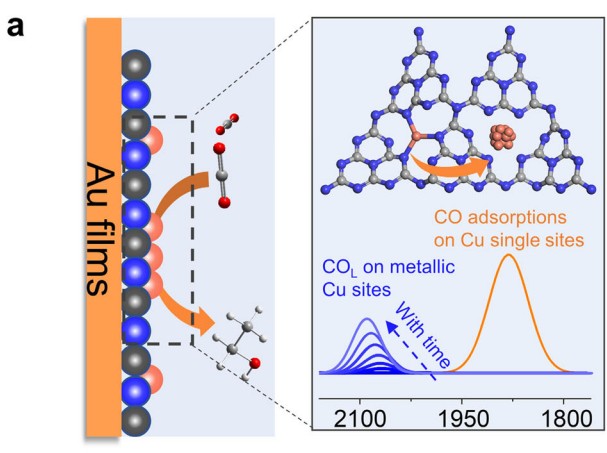

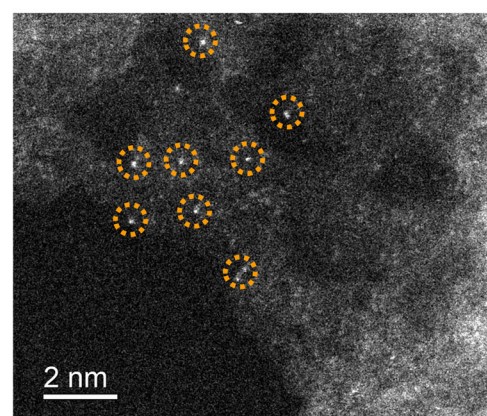

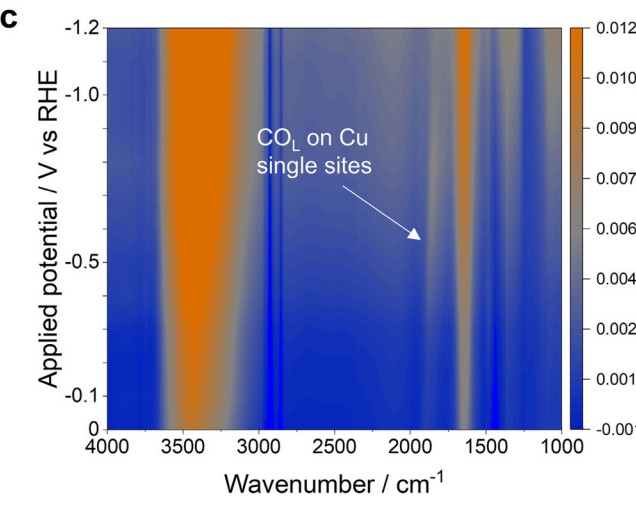

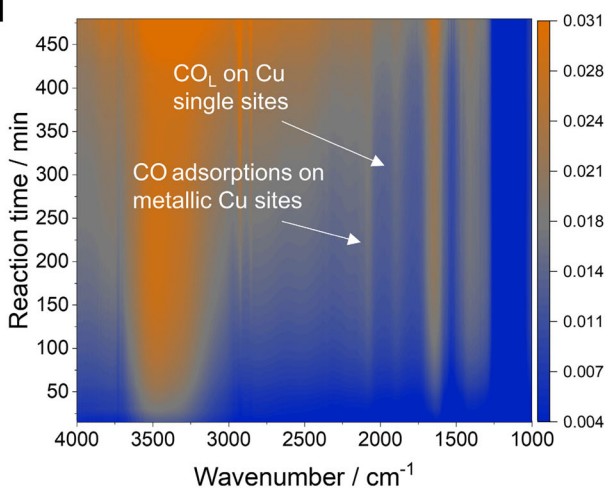

**Fig. 1 | Quantifying the structural evolution of $Cu/C_3N_4$ SACs. a** Schematic illustration showing the reconstruction of $Cu/C_3N_4$ SACs during the $CO_2$RR. **b** HAADF-STEM image of $Cu/C_3N_4$ SACs (selected single Cu atoms are marked by the orange circles). **c** Potential-dependent SEIRA spectra collected in the CO-saturated 0.5 M KHCO$_3$ solution. **d** Time-dependent SEIRA spectra collected in the presence of $CO_2$-saturated 0.5 M KHCO$_3$ solution at −1.2 V.

## Results

### Characterizations of Cu/C₃N₄ SACs

The synthesis of Cu/C₃N₄ SACs follows the previously reported protocols[19]. Scanning electron microscopy (SEM) results show a flower-like structure of the Cu/C₃N₄ catalysts (Supplementary Fig. 1). Transmission electron microscopy (TEM) images show that there are no Cu nanoparticles in the catalysts, indicating that Cu species exist as isolated sites in the Cu/C₃N₄ catalysts (Supplementary Fig. 2). The high-angle annular dark-field scanning transmission electron microscopy (HAADF-STEM) images and energy-dispersive X-ray spectroscopy (EDX) elemental mapping further confirm the existence of isolated Cu sites (marked by orange circles) but no nanoparticles in the Cu/C₃N₄ catalysts (Fig. 1b and Supplementary Fig. 3). Furthermore, the Cu content in Cu/C₃N₄ catalysts is determined to be 0.47 wt% based on the inductively coupled plasma optical emission spectroscopy (ICP-OES) analysis (Supplementary Table 1).

The coordination environment of Cu sites is further investigated by synchrotron-radiation-based X-ray absorption fine structure (XAFS). The X-ray absorption near-edge structure (XANES) spectra show that the absorption energy of Cu in the Cu/C₃N₄ catalysts locates between those in the Cu₂O and CuO catalysts, indicating that the valence state of Cu in the Cu/C₃N₄ catalysts is between +1 and +2 (Supplementary Fig. 4a). The Fourier transform extended X-ray absorption fine structure (FT-EXAFS) of the Cu/C₃N₄ catalysts show a peak at 1.48 Å, which corresponds to the Cu−N and/or Cu−O bonds in the catalysts (Supplementary Fig. 4b)[19,20]. No obvious Cu−Cu bond is observed in the EXAFS spectra, which is consistent to the HAADF-STEM results and further confirms atomic dispersion of Cu species in the

Cu/C₃N₄ catalysts. The FT-EXAFS fitting profile of the Cu K-edge peak indicates that the Cu sites in the Cu/C₃N₄ catalysts are coordinated by three N atoms (Supplementary Fig. 5 and Table 2). Four possible geometrical structures of Cu/C₃N₄ catalysts including Cu-N₃, Cu-N₂OH, Cu-N₂C, and Cu-N₄ are investigated using DFT calculations (Supplementary Fig. 6). It is noted that the calculated XANES spectra of the optimized Cu-N₃ structure are in good agreement with the experimental results (Supplementary Fig. 4a and Fig. 6). In contrast, there is a large discrepancy in terms of peak position and intensity between the calculated XANES spectra of other optimized geometrical structures and experimental results (Supplementary Fig. 4a and Fig. 6). Therefore, Cu-N₃ coordination is likely the geometrical structure of the Cu/C₃N₄ catalysts.

### The structural evolution of Cu/C₃N₄ SACs

*Operando* ATR-SEIRAS is carried out in a customized electrochemical cell to monitor the reconstruction of Cu sites during CO₂RR (Fig. 1a and Supplementary Fig. 7). SEIRA spectra collected on the C₃N₄ substrates show that there is no peak in the range from 1800 cm⁻¹ to 2100 cm⁻¹, suggesting that there is no CO adsorption on the C or N sites of the C₃N₄ substrates (Supplementary Fig. 8). Therefore, the band at around 1890 cm⁻¹ is attributed to linearly bonded CO (CO_L) adsorbed on the Cu single sites of Cu/C₃N₄ catalysts (Fig. 1a, c). It is noted that the peak position of CO_L on Cu single sites is different from those on Cu step and terrace sites (Supplementary Fig. 9). DFT calculations based on the Blyholder model show that the CO adsorption energy is lower on Cu single sites than that on metallic Cu sites (Supplementary Fig. 10a), which leads to the red shift of the CO vibration frequency on Cu single

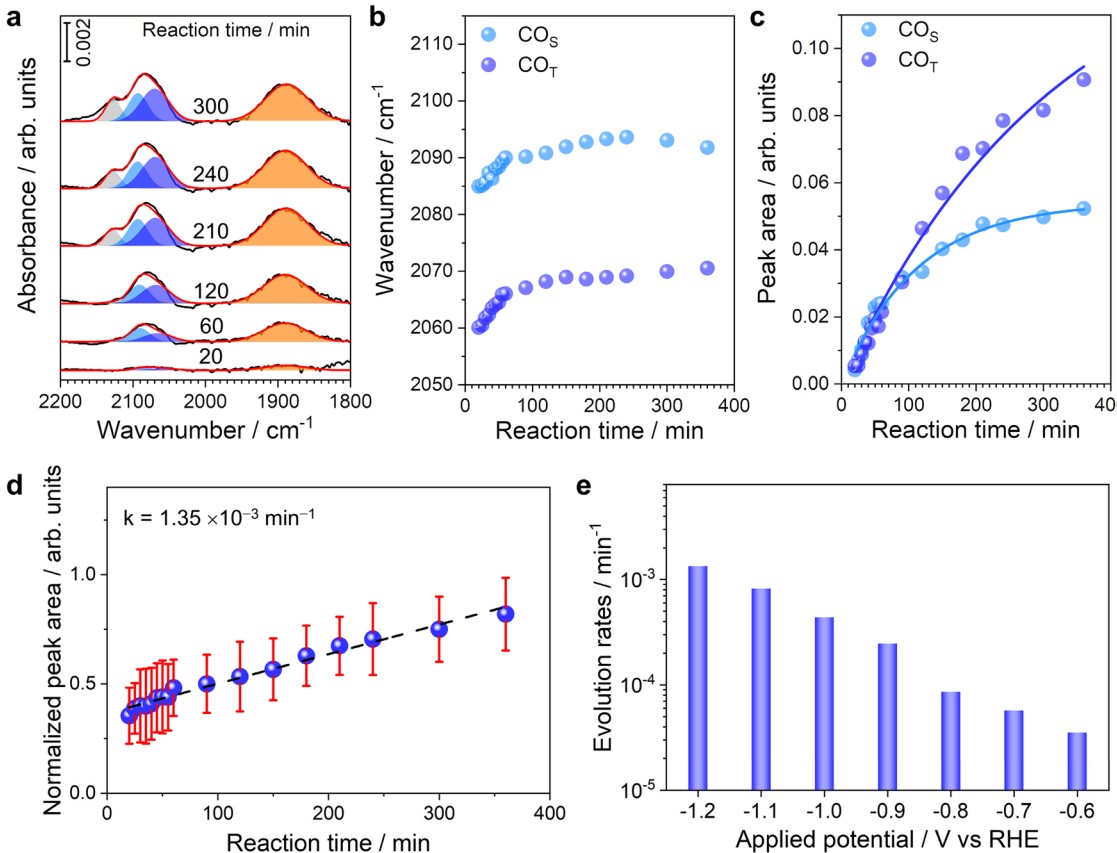

**Fig. 2 | *Operando* SEIRA spectra of Cu/C₃N₄ during the CO₂RR. a** Gaussian fitting of four CO adsorption modes are shown in gray (CO interacting with K cationic species), sky blue (CO adsorption on Cu step sites), blue (CO adsorption on Cu terrace sites), and orange (CO adsorption on Cu single sites), respectively. **b** Frequency plot of changes in the CO adsorptions on metallic Cu sites in *operando*

SEIRA spectra. **c** Peak area plot of changes in the CO adsorptions on metallic Cu sites in *operando* SEIRA spectra. **d** Time-dependent normalized peak area of CO adsorptions on metallic Cu sites at −1.2 V. **e** Potential-dependent evolution rates of Cu single sites. Error bars in **d** represent s.d. for each data point (*n* = 3 independent experiments), and points are average values.

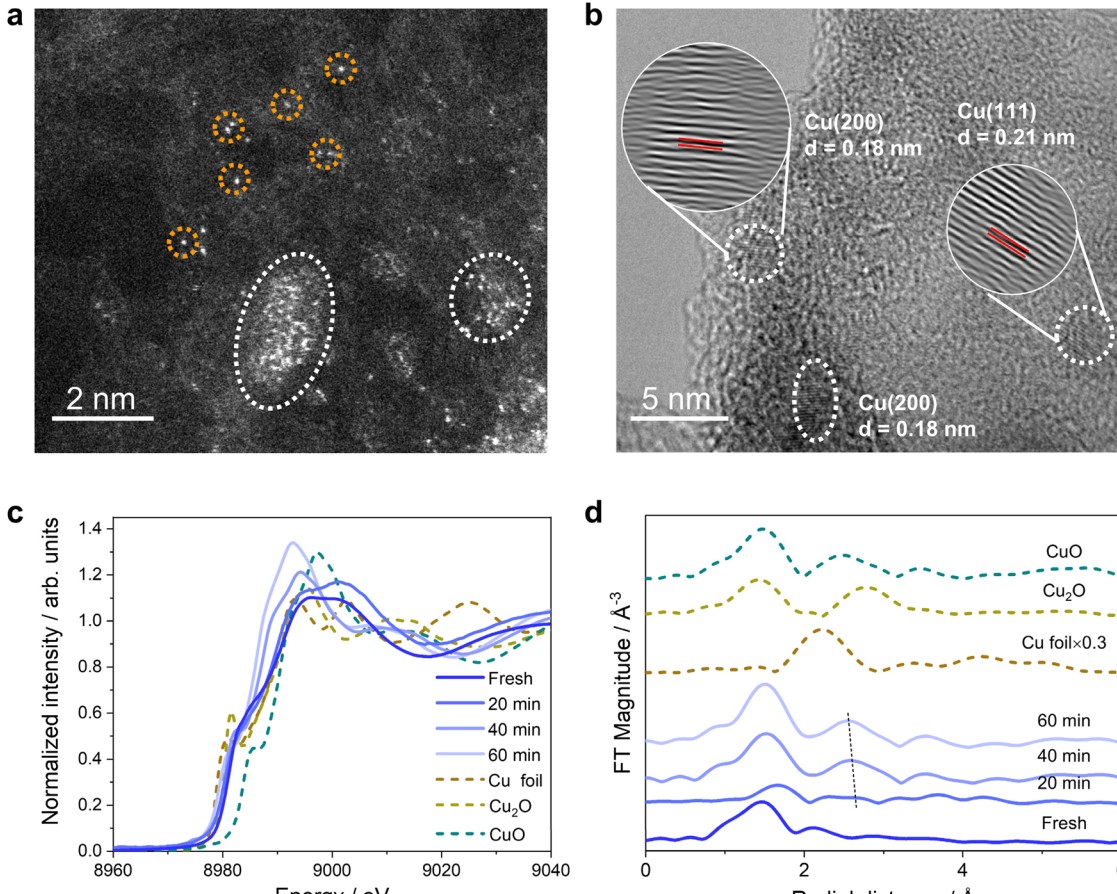

**Fig. 3 | Morphology and structure characterizations of the Cu/C₃N₄ SACs post CO₂RR. a** HAADF-STEM image of Cu/C₃N₄ SACs post CO₂RR at −1.2 V for 8 h (selected single Cu atoms and nanoparticles are marked by the orange circles and white ellipse, respectively). **b** HRTEM images of Cu/C₃N₄ SACs post CO₂RR at −1.2 V showing the lattice fringes corresponding to the Cu(200) and Cu(111) facets. **c** Time-dependence in situ Cu K-edge XANES spectra of Cu/C₃N₄ SACs during the CO₂RR at −1.2 V. **d** Time-dependence in situ Cu K edge FT-EXAFS spectra of Cu/C₃N₄ SACs during the CO₂RR at −1.2 V.

sites compared with that on the step/terrace sites (Supplementary Fig. 10)[21]. The distinct peak position of CO adsorption on Cu single sites from that on metallic Cu sites provide the basis for the spectroscopic quantification of Cu SACs evolutions. The red shift of the $CO_L$ band on Cu single sites with decreasing potentials confirmed its specific adsorption on the catalysts according to the Stark effect (Fig. 1c)[22,23]. Time-dependent SEIRA spectra in $CO_2$-saturated 0.5 M KHCO₃ at −1.2 V vs RHE show that the intensity of $CO_L$ band on Cu single sites increases initially and then reaches a plateau after the CO₂RR for 180 min (Fig. 1d and Supplementary Fig. 11), which could be due to the continuous production of CO during the CO₂RR and limited mass transfer of CO to the Cu sites on the catalysts. No obvious change of peak intensity after 180 min indicates that the amount of Cu single sites is relatively stable during the CO₂RR. The peak position of $CO_L$ on Cu single sites changes negligibly, which is likely due to the lack of interactions between adsorbed CO molecules[24]. It is noted that a new band corresponding to the CO adsorption on metallic Cu sites appears at around 2080 cm⁻¹ after the CO₂RR at −1.2 V for 20 min (Fig. 1d), indicating that Cu single sites in the Cu/C₃N₄ SACs evolve into metallic Cu nanoparticles during the CO₂RR[25,26]. According to previous reports[27,28], the detection limit of X-ray diffraction technique is around 5 nm. If there is significant conversion of Cu single sites into large Cu nanoparticles, diffraction peaks corresponding to Cu nanoparticles would show up in the X-ray diffraction patterns. Ex-situ X-ray diffraction patterns show that the bulk crystal structure of Cu/C₃N₄ catalysts remains the same after being sprayed onto carbon paper and after the CO₂RR for 8 h (Supplementary Fig. 12), demonstrating that there are no significant aggregations

of Cu single sites into large Cu nanoparticles. The intensity of $CO_L$ band on Cu single sites increases until 50 min at −1.2 V in the CO-saturated 0.5 M KHCO₃ solution, which confirms the slow mass transfer of CO to the Cu sites on the catalysts (Supplementary Fig. 13).

Gaussian fitting of *operando* SEIRA spectra shows that the CO adsorption at around 2080 cm⁻¹ can be resolved into three distinct components (Fig. 2a), corresponding to CO adsorptions on three different Cu sites. The main component (sky blue) and low wavenumber component (blue) are associated with CO adsorptions on the Cu step ($CO_S$, -2090 cm⁻¹) and terrace ($CO_T$, -2065 cm⁻¹) sites, respectively. It is noted that the high wavenumber component (gray, -2125 cm⁻¹) appears after CO₂RR at −1.2 V for 20 min. Currently, the assignment of the high wavenumber component is still under debate[26,29–31]. To elucidate the origin of the high wavenumber component, control experiments are conducted with a gold film in CO-saturated 0.05 M KOH and 0.5 M KHCO₃ (Supplementary Fig. 14). The band at -2125 cm⁻¹ appear on the Au film, suggesting that it is corresponding to the CO vibration interacting with K cationic species in the electrical double layer. Further investigations with Cu films in CO-saturated 0.05 M KHCO₃, 0.5 M KHCO₃, and 0.05 M KOH confirm that this band corresponds to CO vibration interacting with K cationic species (Supplementary Fig. 15)[30,31]. The blue shift of CO adsorptions on metallic Cu sites (orange) with reaction time is likely due to the coverage effect (Fig. 2a), which is further confirmed by the increasing intensity of CO adsorptions (Fig. 2c and Supplementary Fig. 16). Considering that produced CO reaches saturation at/near the electrochemical interfaces after 180 min, the continuously increasing CO adsorptions on

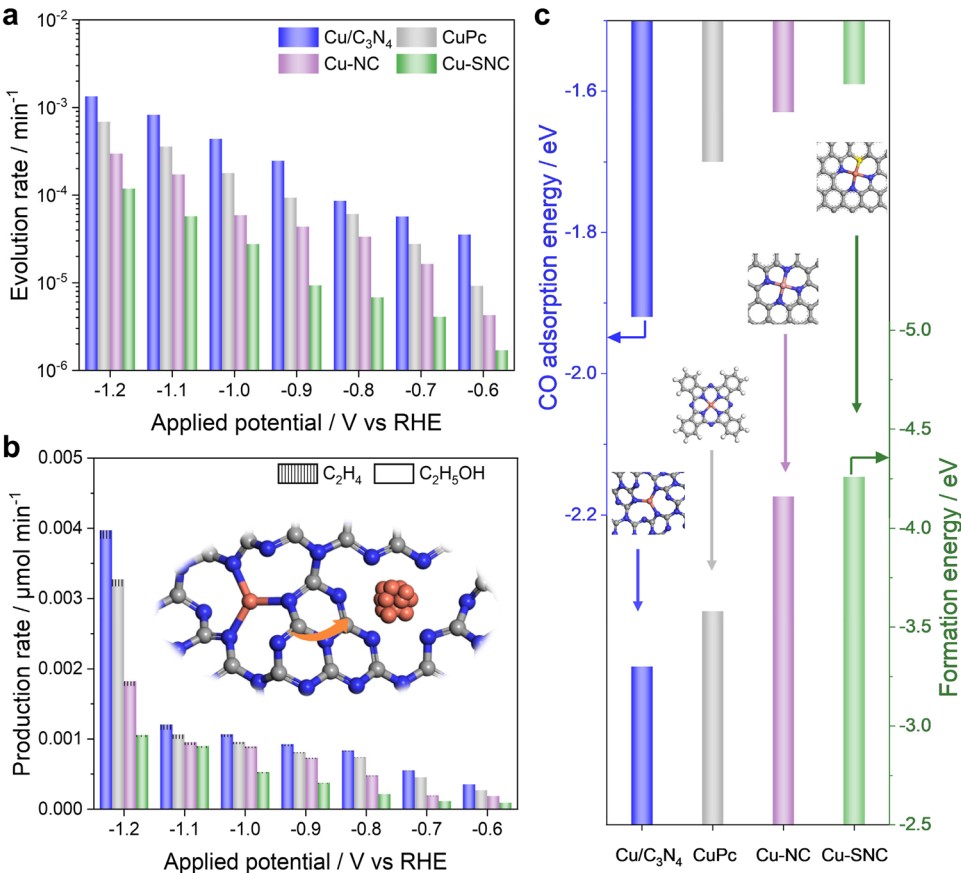

**Fig. 4 | Structure-stability relations of Cu SACs. a** Potential-dependent evolution rates of Cu single sites to metallic Cu sites on Cu/C$_3$N$_4$, CuPc, Cu-NC, and Cu-SNC SACs. **b** Potential-dependent production rates of C$_2$H$_4$ and C$_2$H$_5$OH on Cu/C$_3$N$_4$, CuPc, Cu-NC, and Cu-SNC SACs. **c** DFT calculations of CO adsorption energy and formation energy for Cu/C$_3$N$_4$, CuPc, Cu-NC, and Cu-SNC SACs.

metallic Cu sites is merely due to more metallic Cu sites formed through the reconstruction process during CO$_2$RR (Fig. 2a, b). Since the amount of Cu single sites is relatively stable during the CO$_2$RR, the CO$_L$ on Cu single sites serves as the internal standard for the quantification of CO adsorptions on metallic Cu sites. The evolution rate of Cu single sites to metallic Cu sites on Cu/C$_3$N$_4$ catalysts at −1.2 V vs RHE during the CO$_2$RR is determined to be $1.35 \times 10^{-3}$ min$^{-1}$ (Fig. 2d). Procedures for the calculations of evolution rate are detailed in Supporting Information (section 5). It is found that the evolution rate of Cu single sites is highly dependent on the applied potentials during the CO$_2$RR (Fig. 2e, Supplementary Figs. 17 and 18). The reconstruction of Cu single sites become much more severe with decreasing potentials, specifically, the formation rate of metallic Cu sites at −1.2 V is around two orders of magnitude higher than that at −0.6 V (Fig. 2e).

HAADF-STEM images of the Cu/C$_3$N$_4$ SACs post reaction show that there are Cu nanoparticles formed after CO$_2$RR at −1.2 V for 8 h (Fig. 3a and Supplementary Fig. 19), which is consistent to the SEIRAS results of CO adsorptions on metallic Cu sites. High-resolution transmission electron microscopy (HRTEM) image reveals that the size of as-formed Cu nanoparticles is around 2 nm (Fig. 3b), which shows negligible effect on the SEIRAS according to the FDTD simulations (Supplementary Fig. 20). The lattice distances of 0.18 and 0.21 nm are corresponding to the Cu(200) and Cu(111) facets[32,33], which are in good agreements with the CO adsorptions on Cu step and terrace sites, respectively[34–37]. EDX elemental mapping shows that Cu is evenly distributed in the Cu/C$_3$N$_4$ catalysts and there is no severe aggregation of Cu nanoparticles after CO$_2$RR, indicating that the reconstruction of Cu single sites is minor (Supplementary Fig. 21). In situ XAS studies further confirm the evolution of Cu single sites to metallic Cu sites during

CO$_2$RR. The Cu pre-edge peak shifts to lower energy region during the CO$_2$RR at −1.2 V, suggesting that high-valent Cu single sites are gradually converted to low-valent metallic Cu sites (Fig. 3c and Supplementary Fig. 22). First derivative spectra of normalized Cu K-edge XANES show that both the pre-edge peak and K-edge peak of Cu/C$_3$N$_4$ SACs shift to lower energy region, which further confirms the formation of metallic Cu sites during the CO$_2$RR (Supplementary Fig. 23). An addition peak at -2.6 Å appears after the CO$_2$RR for 20 min in the FT-EXAFS spectra(Fig. 3d), which is attributed to the scattering feature of Cu-Cu interaction[20]. The intensity of the peak corresponding to Cu-Cu interaction slightly increases with time, demonstrating that the conversion of Cu single sites to metallic Cu sites is minor. Therefore, combined ex situ microscopic characterizations and in situ XAS measurements confirm that Cu single sites in the Cu/C$_3$N$_4$ catalysts evolve into Cu nanoparticles during the CO$_2$RR, which are consistent to the *operando* SEIRAS results (Figs. 1d and 2a).

### Structure-stability relations of Cu SACs

*Operando* SEIRAS is further applied to quantitatively investigate the structure-stability relations of other Cu SACs on different substrates during the CO$_2$RR. The syntheses of Cu-NC and Cu-SNC SACs follow the previously reported protocols[38]. SEM images show a rodlike structure of commercial CuPc catalysts and an irregulated hollow structure of Cu-NC and Cu-SNC SACs, respectively (Supplementary Fig. 24). TEM images and EDX elemental mapping show that Cu species are evenly distributed and there are no Cu nanoparticles in the three Cu SACs (Supplementary Figs. 25–27). HAADF-STEM images further confirm that Cu species exist as isolated sites in the catalysts (Supplementary Fig. 28). The coordination environment of the Cu sites in

the three SACs are investigated by XAFS (Supplementary Fig. 29). No obvious Cu−Cu bond is observed in the EXAFS spectra, which is consistent to the HAADF-STEM results and further confirms atomic dispersion of Cu species in the three Cu SACs. The FT-EXAFS fitting profiles of the Cu K-edge peak indicate that the Cu sites in the CuPc, Cu-NC, and Cu-SNC SACs are coordinated by four N atoms ($CuN_4$), four N atoms ($CuN_4$), and one S atom and three N atoms ($CuSN_3$), respectively (Supplementary Fig. 29 and Table 2). Time-dependence SEIRA spectra show that CO adsorption on metallic Cu sites is observed after the $CO_2RR$ at −1.2 V for 60, 165, and 225 min on the CuPc, Cu-NC, and Cu-SNC catalysts, respectively, indicating that the stability of Cu SACs follows the trend that Cu-SNC > Cu-NC > CuPc > $Cu/C_3N_4$ (Supplementary Figs. 30−32). HAADF-STEM images of spent Cu SACs further confirm the formation of Cu nanoparticles post $CO_2RR$ (Supplementary Fig. 33), which is consistent with the spectroscopic observations of CO adsorption on metallic Cu sites. Furthermore, the evolution rates of Cu single sites to metallic sites on different Cu SACs during the $CO_2RR$ are quantified based on the SEIRAS results (Fig. 4a and Supplementary Figs. 34−36). The evolution rates of CuPc, Cu-NC, and Cu-SNC catalysts at −1.2 V are determined to be $6.87 \times 10^{-4}$ $min^{-1}$, $2.97 \times 10^{-4}$ $min^{-1}$, and $1.18 \times 10^{-4}$ $min^{-1}$, respectively. Among the four Cu SACs, $Cu/C_3N_4$ catalysts are the least stable, which is likely due to the low coordination of Cu single sites ($Cu-N_3$). High coordination of the Cu single sites is beneficial to improving the stability of the CuPc and Cu-NC SACs during the $CO_2RR$ ($Cu-N_4$). The evolution rate of Cu-SNC SACs is around one to two orders of magnitude lower than those on the other three catalysts, which is likely due to the strong affinity between S atom and Cu sites[39,40].

To elucidate the active sites for the formation of $C_{2+}$ products, the $CO_2RR$ performance of the Cu SACs are investigated. There are no $C_{2+}$ products but only $H_2$ and CO formed within the first 15 min at −1.2 V for all the four SACs, with the total FEs of $H_2$ and CO as high as -100% (Supplementary Fig. 37). It is noted that $C_{2+}$ products including $C_2H_4$ and $C_2H_5OH$ start to form after the $CO_2RR$ for 30, 60, 200, and 300 min on the $Cu/C_3N_4$, CuPc, Cu-NC, and Cu-SNC SACs, respectively (Supplementary Figs. 38−47), which is in good agreement with the structural evolution of the catalysts as revealed in SEIRAS results (Figs. 1d and 2a). The production rates of $C_{2+}$ products on the Cu SACs follow the trend that $Cu/C_3N_4$ > CuPc > Cu-NC > Cu-SNC (Fig. 4b and Supplementary Fig. 48), which is consistent to the stability of the Cu SACs. Combined microscopic, spectroscopic, and reactivity measurements suggest that metallic Cu sites are the active phase for $C_{2+}$ products via C-C coupling and Cu single sites can only convert $CO_2$ to $C_1$ products.

The DFT calculations are performed to gain insight into the structure-stability relations of Cu SACs. The models of Cu SACs are constructed and optimized based on their crystal structures (Supplementary Fig. 49). Simulated infrared spectra of CO adsorption show that obvious peaks corresponding to $CO_L$ on metallic Cu sites and Cu single sites are located at around 2063 $cm^{-1}$ and 1964 $cm^{-1}$, respectively, which are consistent to the SEIRAS measurements and previous reports (Supplementary Figs. 50 and 51)[36,41]. The calculated CO adsorption energy on the Cu single sites of $Cu/C_3N_4$ SACs show the lowest value of −1.92 eV, indicating that the binding of CO on $Cu/C_3N_4$ SACs is the strongest (Fig. 4c). SEIRAS results show that CO adsorption disappear on the $Cu/C_3N_4$ SACs at a potential higher than 1.3 V, indicating that CO is completely oxidized (Supplementary Fig. 52a). Meanwhile, CO adsorption on the CuPc, Cu-NC, and Cu-SNC catalysts disappear at a potential higher than 1.15, 1.0, 0.95 V, respectively (Supplementary Fig. 52b-d). Therefore, the active Cu singles sites on $Cu/C_3N_4$ SACs are likely blocked by the intermediate *CO during the $CO_2RR$ due to its strong adsorption, leading to a low FE of CO compared with other Cu SACs[42]. The calculated formation energy of the Cu SACs follows the trend that $Cu/C_3N_4$ (−3.30 eV) > CuPc (−3.58 eV) > Cu-NC (−4.16 eV) > Cu-SNC (−4.26 eV), which is

consistent to the stability of Cu SACs from SEIRAS and reactivity results. The adsorption of H is revealed to be a vital driving force for the leaching of Cu single sites from the catalyst surfaces (Supplementary Fig. 53). The adsorption of H on the Cu SACs becomes stronger with the decreasing potentials, leading to the leaching of Cu single sites by weakening the Cu-N bonds. The collision of the Cu atoms forms a transient Cu cluster (Supplementary Fig. 54), which is consistent to previous reports[43]. It is also noted that not only SACs but also nanoparticle catalysts tend to aggregate during electrochemical reactions due to various factors such as reaction intermediate-metal bonding, potential, gas evolution, and cathodic corrosion[44–47], which can significantly impact catalytic performance and stability. In the past few year, different strategies have been developed to achieve improved catalytic performance with high stability, including the synthesis of electrocatalysts with different shapes, compositions and structures, coating the electrocatalysts with ultrathin carbon shells, and etc[48,49]. Combined DFT calculations and experimental results demonstrate that the coordination environment is of great importance in tailoring the stability of Cu SACs. Therefore, the stability of Cu SACs can be manipulated by controlling the coordination environment of Cu single sites.

## Discussion

In summary, this study provides a direct structure-stability relationship of Cu SACs for $CO_2RR$. The catalyst reconstruction that occurs to a variety of Cu SACs including $Cu/C_3N_4$, CuPc, Cu-NC, and Cu-SNC catalysts have been quantitatively investigated under the $CO_2RR$ conditions using *operando* ATR-SEIRAS. Combined microscopic, spectroscopic, and reactivity results reveal that Cu SACs evolves into Cu nanoparticles during the $CO_2RR$ and Cu nanoparticles are the active phase for the $C_{2+}$ products. The evolution rate is highly dependent on the applied potentials and the affinity between the Cu single sites and catalyst substrates. The evolution rate of Cu SACs increases rapidly with decreasing potentials, specifically, the formation rate of metallic Cu sites at −1.2 V is around two orders of magnitude higher than that at −0.6 V. By introducing a strong coordinate S and increasing the Cu coordination number simultaneously, as-synthesized Cu-SNC catalysts show significantly enhanced structure stability compared to the other three Cu SACs. DFT calculations demonstrate that the coordination environment is highly important in tailoring the stability of Cu SACs. However, the relatively poor sensitivity and reproducibility of chemically deposited metal films make it challenging to achieve quantitative measurements of reaction kinetics using ATR-SEIRAS technique. We believe that the fabrication of SEIRAS substrates with uniform size and well-defined morphology would be a promising direction for achieving quantitative understanding of reaction mechanisms. Taken together, this work offers an attractive strategy to achieve mechanistic understandings of structure-stability relationship using *operando* ATR-SEIRAS.

## Methods

### Chemicals and materials

Melamine (99%), cyanuric acid (99%), copper(II) nitrate trihydrate (99%), ammonium fluoride (98%), ammonium chloride (99.5%), sodium thiosulfate pentahydrate (99.99%), potassium bicarbonate (99.7%), 40% hydrofluoric acid, copper(II) phthalocyanine (99%), copper chloride dihydrate (99%), sodium chloride (99%), Nafion (5 wt%), copper sulfate pentahydrate (99%), sulfur powder (99.5%), ethylenediaminetetraacetic acid disodium salt ($Na_2EDTA$, 98%), and copper(II) phthalocyanine (CuPc, 99%) were purchased from Shanghai Aladdin Biochemical Technology. Dimethyl sulfoxide (DMSO, 99.5%), formaldehyde (37–40%), ethanol (99.7%), isopropyl alcohol (IPA, 99.7%), and carbon tetrachloride (99.5%) were obtained from Sinopharm Chemical Reagent Co., Ltd. Glucose (99%) and deuterium oxide ($D_2O$,

99.9%) was purchased from Shanghai Macklin Biochemical Co., Ltd. Carbon dioxide (99.999%), carbon monoxide (99.999%), nitrogen (99.999%), and argon (99.999%) were purchased from Wuhan Zhongxin Ruiyuan Gas Co., Ltd.

## Preparation of Cu single-atom catalysts

Cu/$C_3N_4$ SACs were synthesized following a "one pot" method[19]. In a standard synthesis, 0.50 g of melamine and 0.02 g of $Cu(NO_3)_2 \cdot 3H_2O$ were dissolved in 20 mL of DMSO under ultrasonication for 30 min. 0.51 g of cyanuric acid was dissolved in 10 mL of DMSO under ultrasonication for 30 min. At first, 10 mL of cyanuric acid solution was added into 20 mL of melamine and $Cu(NO_3)_2 \cdot 3H_2O$ solution under magnetic stirring for 30 min at room temperature. The mixed solution would turn to blue temporarily and then white precipitate formed. Subsequently, the white precipitate was dried off at 70 °C for 12 h after washing with DI-water and ethanol three times. Finally, the dried powder was pyrolyzed at 550 °C under $N_2$ atmosphere for 4 h at a ramp rate of 2.3 °C min⁻¹.

Cu-NC and Cu-SNC were synthesized according to previous reports[38]. Firstly, 4.387 g of NaCl, 1 mL of 10 mg mL⁻¹ $CuCl_2 \cdot 2H_2O$, and 0.416 g of glucose were added into 4 mL of DI-$H_2O$ under ultrasonication for 1 h. Obtained homogeneous mixture was dried off by freeze-drying for 48 h. Then, the obtained powder was nitrided under ammonia atmosphere at 500 °C for 4 h at a ramp rate of 5 °C min⁻¹. The calcined powder was dispersed in 1000 mL of DI-$H_2O$ and stirred for 12 h to remove extra NaCl. Subsequently, the cleaned powder was calcined under Ar atmosphere at 900 °C for 4 h at a ramp rate of 5 °C min⁻¹. The Cu-NC SACs were obtained after acid-washed with 1 M HCl solution at 80 °C for 12 h to removed Cu nanoparticles. In a standard synthesis of Cu-SNC SACs, 5 mg of sulfur powder and 10 mg of Cu-NC SACs were dissolved in a mixture of 80% carbon tetrachloride and 20% ethanol under ultrasonication for 2 h. After evaporating the solvent under magnetic stirring at 60 °C, the obtained powder was calcined under Ar atmosphere at 450 °C for 2 h and then maintained at 900 °C for another 4 h. The Cu-SNC SACs were obtained after acid-washed with 1 M HCl solution at 80 °C for 12 h to removed Cu nanoparticles.

## Preparation of Au films

Au film electrodes were chemically deposited on Si prisms according to previous reports[50]. Typically, the prism was first polished with a 0.05 μm $Al_2O_3$ slurry and sonicated in acetone and water to remove the residue. Following cleaning, the reflecting plane of the prism was immersed in 40% $NH_4F$ for 2 min to remove the oxide layer and create a hydrogen-terminated surface. The Si surface was then immersed in a 4.4:1 by volume mixture of 2% HF and Au plating solution consisting of 5.75 mM $NaAuCl_4 \cdot 2H_2O$, 0.025 M $NH_4Cl$, 0.075 M $Na_2SO_3$, 0.025 M $Na_2S_2O_3 \cdot 5H_2O$, and 0.026 M NaOH at 55 °C for 5 min. Finally, the Au films were washed with DI-$H_2O$ for further use.

## Preparation of Cu films

Cu film electrodes were chemically deposited on Si prism according to previous reports[26]. In a standard synthesis, the prism was first polished with a 0.05 μm $Al_2O_3$ slurry and sonicated in acetone and water to remove the residue. Following cleaning, the reflecting plane of the prism was immersed in 40% $NH_4F$ for 2 min to remove the oxide layer and create a hydrogen-terminated surface. The Si surface was then immersed in a Cu plating solution (pH = 12.2) consisting of 0.25 M HCHO, 0.02 M $CuSO_4$, and 20 mM $Na_2EDTA$, 0.3 mM $Na_2S_2O_3 \cdot 5H_2O$ at 55 °C for 5 min. Finally, the Cu films were washed with DI-$H_2O$ for further use. The Cu films were activated at −0.5 V for 10 min and the reference spectrum was collected at open current potential under Ar atmosphere.

## *Operando* ATR-SEIRAS measurements

*Operando* ATR-SEIRAS experiments were conducted in a customized spectroelectrochemical cell reported in our previous works (Supplementary Fig. 6)[51]. Typically, 1 mg of the catalysts was dispersed in a mixture of 1.92 mL of DI-$H_2O$, 1.92 mL of IPA, and 160 μL of 5 wt% Nafion under ultrasonication for 1 h to produce an ink with a catalyst concentration of 0.25 mg mL⁻¹. 200 μL of the ink was then placed on an Au film, which was utilized as the working electrode. An Ag/AgCl electrode (Gaoss Union) and a graphite rod were used as the reference electrode and counter electrode, respectively. The electrolyte was purified through an electrolysis process by maintaining a constant negative current on an Au foil working electrode to extract any potential metal impurities. All spectroscopic measurements were collected with 4 cm⁻¹ resolution and at least 128 coadded scans using a Thermo Fisher Nicolet iS50 FTIR spectrometer equipped with a liquid nitrogen-cooled MCT detector. Electrochemical measurements were conducted using a VersaSTAT 3 potentiostat galvanostat. Impedance measurements were conducted at the beginning of each experiment, and the internal resistance (typically ~40 Ω) was actively corrected for throughout all experiments. Reference spectra were collected at 1.0 V vs RHE in $CO_2$-saturated 0.5 M $KHCO_3$ solution. All reported potentials in this work are referenced to the RHE unless noted otherwise.

## Characterization of catalysts

The morphology of catalysts was characterized by scanning electron microscopy (SEM, Gemini SEM 300, Carl Zeiss) and transmission electron microscopy (TEM, JEM-F200, JEOL). High-angle annular dark-field canning transmission electron microscopy (HAADF-STEM) imaging was performed on a JEM-ARM200F microscope (JEOL). The Cu contents in Cu SACs were measured using inductively coupled plasma optical emission spectrometry (ICP-OES, Optima 7300 DV, PerkinElmer). X-ray diffraction (XRD) patterns were collected on a Rigaku SmartLab SE automated multipurpose X-ray diffractometer (Japan). The microscopic characterizations of spent catalysts were conducted after the $CO_2$RR at −1.2 V for 8 h.

## Ex situ and in situ XAS measurements

The XAS experiments were conducted at BL14W1 beamline of Shanghai Synchrotron Radiation Facility (SSRF) at room temperature. A double Si (111)-crystal monochromator was used for energy selection. Energy calibration was performed with a Cu foil standard by shifting all spectra to a glitch in the incident intensity. Fluorescence spectra were recorded using a seven-element Ge solid state detector. The acquired XAS results were analyzed by the Demeter software packages. The amplitude reduction factor $S_0^2$ was obtained using Cu foil as the reference.

In situ Cu K-edge XANES spectra of Cu/$C_3N_4$ SACs were collected at −1.2 V at BL14W1 beamline of SSRF at room temperature. The ink solution of Cu/$C_3N_4$ SACs was hand-sprayed onto carbon paper (Gaoss Union) to an approximate loading of 2.5 mg cm⁻² to serve as the working electrode. During the in situ XAS measurements, $CO_2$ gas was introduced into the 0.5 M $KHCO_3$ solution at a flow rate of 20 mL min⁻¹. It should be noted that the bubbles produced during the introduction of $CO_2$ gas would interfere with the XAS measurements. Low flow rate of $CO_2$ gas was critical to in situ XAS measurements.

## Electrochemical measurements

Electrochemical experiments were carried out in an H-type electrochemical cell separated by a Nafion-117 membrane using a CHI 760 potentiostat. The ink solution of Cu SACs was hand-sprayed onto carbon paper (Gaoss Union) to an approximate loading of 0.05 mg cm⁻² to serve as the working electrode. The area of working electrode was fixed to be 0.5 cm² and the total amount of Cu SACs on carbon paper was almost the same as that on Au films during *operando* SEIRAS measurements. An Ag/AgCl electrode (Gaoss Union) and a graphite rod were used as the reference electrode and counter electrode, respectively. The electrolyte was purified through an electrolysis process by maintaining a constant negative current on an Au foil working electrode to extract any potential metal impurities. The

reactivity tests for the $CO_2RR$ were conducted in $CO_2$-saturated 0.5 M $KHCO_3$ solution. The gaseous products including CO, $H_2$, and $C_2H_4$ were quantified using a gas chromatograph (Agilent 8890). Liquid products such as $C_2H_5OH$ were quantified using a nuclear magnetic resonance (NMR) spectrometer (Bruker AVANCE NEO 600 MHz). Typically, 0.5 mL of the electrolyte was taken out and mixed with 0.1 mL of $D_2O$ containing 0.1 ppm DMSO for the $^1H$ NMR measurements.

## DFT calculations

Density functional theory (DFT) calculations including geometry optimizations and frequency calculations were performed using a Dmol$^3$ module of Material Studio 2020. The generalized gradient approximation (GGA) method with Perdew–Burke–Ernzerhof (PBE) function was employed to describe the interactions between core and electrons[52,53]. The force and energy convergence criterion were set to 0.002 Ha Å$^{-1}$ and $10^{-5}$ Ha, respectively. The adsorption energy ($\Delta E_{ad}$) was calculated as:

$$\Delta E_{ad} = E_{total} - E_{cat} - E_{CO} \tag{1}$$

where the $E_{total}$ is the energy of optimized structure (CO-Cu/$C_3N_4$, CO-CuPc, CO-Cu-NC, and CO-Cu-SNC). $E_{cat}$ is the energy of catalysts (Cu/$C_3N_4$, CuPc, Cu-NC, and Cu-SNC SACs). $E_{CO}$ is the energy of CO molecule.

The formation energy ($\Delta E_F$) was calculated as:

$$\Delta E_F = E_{cat} - E_{sub} - E_{Cu} \tag{2}$$

where the $E_{cat}$ is the energy of optimized Cu SACs (Cu/$C_3N_4$, CuPc, Cu-NC, and Cu-SNC SACs). $E_{sub}$ is the energy of the substrates for Cu/$C_3N_4$ ($C_3N_4$), CuPc (Pc), Cu-NC ($N_4$), and Cu-SNC ($SN_3$) catalysts. $E_{Cu}$ is the energy of a single Cu atom.

## Data availability

All data generated or analyzed during this study are included in the published article and its supplementary information files.

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

## Acknowledgements

L.Z., XiaojuY., Q.Y., Z.Y. and XuanY. acknowledge the financial support from the National Natural Science Foundation of China (Grant. No. 22204054), the HUST Academic Frontier Youth Team grant (Grant No. 2019QYTD11), Knowledge Innovation Program of Wuhan-Shuguang, and Wuhan Talented Youth Program. T.C., C.R., F.-Z.X. and B.Z. thank the support from the National Natural Science Foundation of China (Grant. No. 52105145, No. 12274124) and the Shanghai pilot Program for Basic Research (Grant. No. 22TQ1400100-6). Z.W., J.D., Q.Z. and Y.Z. acknowledge the Center for Electron Microscopy at Wuhan University for their substantial supports to TEM characterizations. Thanks engineer Wei Xu in Optoelectronic Micro&Nano Fabrication and Characterizing Facility, Wuhan National Laboratory for Optoelectronics of Huazhong University of Science and Technology for the support in device fabrication.

## Author contributions

XuanY. conceived and planned the project. L.Z. and XiaojuY. conducted the SEIRAS experiments. L.Z., J.D., Q.Y., Z.W., Q.Z. and Y.Z. performed the TEM and XAFS characterizations. Z.Y. prepared Cu films. T.C., C.R., F.-Z.X. and B.Z. carried out the DFT calculations and FDTD simulations. L.Z., B.Z. and XuanY. analyzed the experimental data and prepared the manuscript. All authors discussed the results and contributed to the manuscript.

## Competing interests

The authors declare no competing interests.
