## [Peer Review File · Nature Communications]

REVIEWER COMMENTS

Reviewer #1 (Remarks to the Author):

This manuscript reports that the catalysts of Cu single-atom on C₃N₄, porphyrin, NC, and SNC can reconstructs to Cu nanoparticles during CO₂ reaction reduction, and the density functional theory is used to calculate the adsorption energy of CO and formation energy of SACs, which used to demonstrate the stability of Cu single atoms structures. Furthermore, the operando ATR-SEIRAS is used to understand the structure-stability relationship. Although the view is relatively innovative and the experimental phenomena are interesting, I would not recommend for publication in Nature Communications. The detailed comments are listed as follows.

1. The geometrical structures of Supplementary Figure 5 (inset) is not same as Figure 1A (right), Figure 4B, and Supplementary Figure 33A. The number of bond length of all Cu-N bond need to increase in Supplementary Figure 33A, because it is not like the CuN₃ structure.
2. The band of at 1890 cm⁻¹ is attributed to the linearly bonded COL, while the frequency of COL of the calculation results at 1970 cm⁻¹, there such a discrepancy between Operando ATR-SEIRAS and DFT simulation, so further investigations are needed to elucidate the origin of this band. Moreover, the authors used the bonding interaction between CO and Cu to explain the frequency different of CO adsorption on Cu single atom and step/terrace sites, but a detail analysis is not, so the bonding interaction analysis is need in the article, such as the Dewar–Chatt–Duncanson model. And the reference 21 which is cited in here, the blue shift of the CO vibration frequency on Pt single atom compared with the Pt clusters (Figure 6 of reference 21), while in this paper, a red shift of the CO vibration frequency on Cu single atom compared with the step/terrace sites.
3. The wavenumber of ~2125 cm⁻¹ is attributed to CO vibration interacting with K cationic species in electrical double layer, the further investigation is also needed to elucidate the origin of this band. Although the reference 26 is cited in here, in the reference 26, the control experiments of KHCO₃ produced by CO₂ saturating KOH on the Cu-poly surface are used to eliminate the role of electrolyte electrochemically deposited on the electrode, and confirming that this band corresponds to CO adsorbed on Cu sites.
4. Line 167-169, the Cu(200) and Cu(111) facets are verified by lattice distances, and two facets used to explain the CO adsorptions on Cu step and terrace sites, respectively. while, the single crystal surface of Cu(200) is same as the Cu(100), both Cu(100) and Cu(111) are regarded as the low-index surfaces, as the reference 29 said, the Cu(211), Cu(221), and Cu(532) are the regular step and kink surfaces.
5. The geometrical structure of CO adsorption on metallic and monoatomic Cu sites Cu/C₃N₄, CuPc, Cu-NC, and Cu-SNC is necessary.
6. Ex-situ X-ray diffraction patterns demonstrate that small portion of Cu single site are converted into metallic Cu sites after CO₂RR for 8h, but the C₂⁺ products start to form after the CO₂RR for 30 min on the Cu/C₃N₄.

7. Although the reconstruction process of Cu SACs convert into Cu nanoparticles is found by using the ATR-SEIRAS, a physical image of adsorbed CO promoting the Cu aggregation is not clear.
8. The TOC is very confusing of the structure of two adjacent CO on Cu nanoparticle.

Reviewer #2 (Remarks to the Author):

The authors conducted a comprehensive investigation on the structural evolution of Cu single-atom catalysts (SACs) during the electrochemical CO₂ reduction. They employed operando infrared spectroscopic measurements to monitor CO adsorption on both single atom and metal cluster sites. According to their findings, the Cu SACs initially generated C₁ product (CO) owing to the absence of C-C coupling. However, as the reaction progressed, they observed the gradual formation of C₂ products due to the segregation of Cu SACs into metallic nanoparticles. Another important observation was that the stability of SACs proved to be significantly influenced by the substrates, attributed to differences in coordination. This work is well organized and the interpretation of experimental results is executed well. Nevertheless, to ensure acceptance, a major revision is recommended to address certain aspects.

1) Elaborate on the segregation process of single-atom catalysts (SACs): Provide a more comprehensive explanation of the process. What drives the breaking of Cu-N coordination bonds and the formation of Cu-Cu metallic bonds?

2) Not only single-atom catalysts (SACs) but also Cu nanoparticle catalysts tend to aggregate during electrochemical reactions due to various factors such as reaction intermediate-metal bonding [Science 351, 475 (2016)], potential [Nat. Commun. 9, 1 (2018)], gas evolution [Nat. Energy 7, 537 (2022)], or cathodic corrosion [ACS Catal. 12, 13174 (2022)]. This broad aspect of catalyst aggregation can significantly impact catalytic performance and stability. To address these challenges, it is highly recommended to discuss potential strategies that can effectively mitigate catalyst aggregation, taking into account the insights gained from your results and also citing the relevant references.

3) The faradaic efficiency of C₂⁺ products is observed to be lower than 1% (Figure S31), and the production rate is also found to be lower than 0.04 μmol/min (Figure S32). These values appear to be very small and may raise concerns about reliability. Additionally, it is amazing that the observed trends of catalytic activity, with Cu/C₃N₄ > CuPc > Cu-NC > Cu-SNC, are quite remarkable, especially given the very small error range (FEs < 0.2%). To ensure the reliability of these results, please provide further evidence, such as the original spectrum of C₂ product peaks in NMR or gas chromatograph data.

Reviewer #3 (Remarks to the Author):

The manuscript by Zhang et al. reported the quantitatively monitoring of the structure evolution of Cu single-atom catalysts during the electrochemical reduction of CO₂. The evolution rate of Cu single-atom catalysts on different substrates were quantified using surface-enhanced infrared absorption spectroscopy (SEIRAS). Combined with DFT calculations, it was found that the stability of Cu SACs is highly dependent on their formation energy, which can be manipulated by controlling the affinity between Cu sites and substrates. I enjoy very much on reading this manuscript, and the method used by the authors might be highly useful for others to study the catalyst stability of single atom catalysts. I would like to recommend the publication of this manuscript after some minor concerns are fully addressed.

As far as I know, it is challenging to use ATR SEIRAS to quantitatively measure the chemical kinetics of the reactions of adsorbates on catalyst surface, especially with various particle size and morphology. I suggest the authors claim the limitations of their method for the calculation of chemical kinetics.

In terms of the kinetic analysis, the authors may want to consider the aggregation of reduced Cu(0) for the formation of Cu nanoparticles. Also, the authors need to clarify that the density of adsorbed CO molecules on the surfaces of Cu nanoparticles is a constant based on their analysis.

For the quantification, will the formed Cu nanoparticles affect the signal of SEIRA spectra, since chemically deposited Cu films consisting of Cu nanoparticles are widely utilized as the substrates for SEIRAS studies?

I would like to suggest the authors to add the proposed structures of all four copper catalysts in the main text, especially for the Cu/SNC catalyst.

For the proposed chemical structure of Cu/C₃N₄, did the authors consider Cu-N₂-OH coordination configuration where Cu is coordinating with the edge NN sites and an OH group? To me, the currently proposed structure of Cu-N₃ is with high structural reorganization, and the coordinating N sites is out of the C₃N₄ plan. Considering the low Cu loading and poor stability of the catalyst, edge NN sites should not be ruled out. DFT calculations should be carried out to address this issue.

Line 132. Read "Ex-situ X-ray diffraction patterns show that the bulk crystal structure of Cu/C₃N₄ catalysts remains the same after being sprayed onto carbon paper and after the CO₂RR for 8 h (Supplementary Figure 9), demonstrating that a small portion of Cu single sites are converted into metallic Cu sites." Amorphous sub-nano Cu clusters could not be ruled out by X-ray diffraction and thereby it is not safe to mention only a small portion of Cu single sites are converted into metallic sites. Especially, Figure 3A displays more metallic copper atoms than single Cu atoms.

Reviewer #1

Recommendation:

“This manuscript reports that the catalysts of Cu single-atom on C_3N_4 , porphyrin, NC, and SNC can reconstructs to Cu nanoparticles during CO_2 reaction reduction, and the density functional theory is used to calculate the adsorption energy of CO and formation energy of SACs, which used to demonstrate the stability of Cu single atoms structures. Furthermore, the operando ATR-SEIRAS is used to understand the structure-stability relationship. Although the view is relatively innovative and the experimental phenomena are interesting, I would not recommend for publication in Nature Communications. The detailed comments are listed as follows.”

Response: We thank the reviewer for his/her comments and have revised our manuscript accordingly.

1) *“The geometrical structures of Supplementary Figure 5 (inset) is not same as Figure 1A (right), Figure 4B, and Supplementary Figure 33A. The number of bond length of all Cu-N bond need to increase in Supplementary Figure 33A, because it is not like the CuN_3 structure.”*

Response: We thank the reviewer for the valuable comments. The geometrical structures of Supplementary Figure 5 (inset) in previous version of the manuscript is the original structure before optimization process of DFT calculations. We are sorry for the confusion and have updated the geometrical structures of Supplementary Figure 5 (inset) to be the same as Figure 1A (right), Figure 4B, and Supplementary Figure 49A in the revised manuscript and supplementary information.

Regarding the bond length of the Cu-N bond in Cu/C_3N_4 catalysts (Supplementary Figure 49A), we have utilized DFT calculations to further optimize the structure of Cu/C_3N_4 catalysts. DFT calculations show that the interaction between Cu and N is too weak to form Cu-N bond if the bond length of Cu-N bond further increases. Furthermore, we have investigated other possible geometrical structures of Cu/C_3N_4 catalysts including Cu- N_2OH , Cu- N_2C , and Cu- N_4 with DFT calculations (Supplementary Figure 6 in the revised supplementary information). It is noted that the calculated XANES spectra of the optimized Cu- N_3 structure are in good agreement with the experimental results (Supplementary Figure 4A and Figure 6 in the revised supplementary information). In contrast, there is a large discrepancy in terms of peak position and intensity between the calculated XANES spectra of other optimized geometrical structures (Cu- N_2OH , Cu- N_2C , and Cu- N_4) and experimental results (Supplementary Figure 4A and Figure 6 in the revised supplementary information). Therefore, we believe that the optimized Cu- N_3 structure in the revised manuscript can represent the geometrical structure of the Cu/C_3N_4 catalysts.

Action: We have updated the geometrical structures of the Cu/C_3N_4 catalysts in the revised manuscript to make sure that the geometrical structures in Supplementary Figure 5 (inset), Figure 1A (right), Figure 4B, and Supplementary Figure 49A are consistent with each other in the revised manuscript and supplementary information.

We have also attached the simulated XANES spectra as Supplementary Figure 6 on Page 12 in the revised supplementary information and added the following sentences on Page 5 in the revised manuscript when discussing the optimization of Cu/C_3N_4 structures:

“Four possible geometrical structures of Cu/C_3N_4 catalysts including Cu- N_3 , Cu- N_2OH , Cu- N_2C , and Cu- N_4 are investigated using DFT calculations (Supplementary Figure 6). It is noted that the calculated XANES

spectra of the optimized Cu-N₃ structure are in good agreement with the experimental results (Supplementary Figure 4A and Figure 6). In contrast, there is a large discrepancy in terms of peak position and intensity between the calculated XANES spectra of other optimized geometrical structures and experimental results (Supplementary Figure 4A and Figure 6). Therefore, Cu-N₃ coordination is likely the geometrical structure of the Cu/C₃N₄ catalysts.”

Supplementary Figure 6. Simulated XANES spectra for four possible geometrical structures of Cu/C₃N₄ catalysts. (A) Cu-N₃. (B) Cu-N₂OH. (C) Cu-N₂C. (D) Cu-N₄.

2) “The band of at 1890 cm⁻¹ is attributed to the linearly bonded CO_L, while the frequency of CO_L of the calculation results at 1970 cm⁻¹, there such a discrepancy between Operando ATR-SEIRAS and DFT simulation, so further investigations are needed to elucidate the origin of this band. Moreover, the authors used the bonding interaction between CO and Cu to explain the frequency different of CO adsorption on Cu single atom and step/terrace sites, but a detail analysis is not, so the bonding interaction analysis is need in the article, such as the Dewar–Chatt–Duncanson model. And the reference 21 which is cited in here, the blue shift of the CO vibration frequency on Pt single atom compared with the Pt clusters (Figure 6 of reference 21), while in this paper, a red shift of the CO vibration frequency on Cu single atom compared with the step/terrace sites.”

Response: We thank the reviewer for the valuable comments. First, we would like to emphasize that the peak position of the band assigned to the linearly bonded CO adsorbed on Cu single sites (CO_L) changes with potential because of the Stark effect (Figure 1C and Supplementary Figure 52). The band locates at

$\sim 1890\text{ cm}^{-1}$ at -1.2 V vs RHE and shifts to high frequency region with the increasing potential. The peak position of the CO_L band shifts to 1985 cm^{-1} at 1.2 V vs RHE, which is very close to the calculated frequency of CO_L at $\sim 1970\text{ cm}^{-1}$ (Supplementary Figure 51A), indicating a high consistency between the experimental and computational results. To elucidate the origin of this band, we further investigated the CO adsorption on C_3N_4 in CO-saturated 0.5 M KHCO_3 using ATR-SEIRAS (Supplementary Figure 8 on Page 14 in the revised supplementary information). There is no peak in the range from 1800 cm^{-1} to 2100 cm^{-1} , suggesting that the band is unlikely corresponding to the CO adsorption on the C_3N_4 substrates. Meanwhile, we have observed the same band located at nearly the same position on the four different Cu single-atom catalysts in CO-saturated 0.5 M KHCO_3 . Therefore, we believe that it is reasonable to attribute the band at $\sim 1890\text{ cm}^{-1}$ at -1.2 V vs RHE to the linearly bonded CO adsorbed on the Cu single sites (CO_L).

We further carried out DFT calculations to analyze the bonding interaction between CO molecules and Cu sites based on the Dewar–Chatt–Duncanson model. However, the CO adsorption on Cu sites turned into Blyholder model during the optimization process (Figure R1 in the response letter), which has been widely utilized to study the CO adsorptions on the surfaces of transition metals (Refs: *J. Phys. Chem.* **1964**, 68, 2772–2777, *Chem. Soc. Rev.* **2010**, 39, 4643–4655). Therefore, the bonding interactions between CO molecules and Cu sites are analyzed based on the Blyholder model (Supplementary Figure 10 on Page 16 in the revised supplementary information). The calculated infrared spectra of CO adsorptions show that the peaks corresponding to CO_L on Cu single sites and metallic Cu sites are located at around 1964 and 2063 cm^{-1} , respectively, which are in good agreement with the SEIRAS results (Figure R2 in the response letter, Supplementary Figure 9 and 52). DFT calculations show that the CO adsorption energy is lower on Cu single sites than that on metallic Cu sites (Supplementary Figure 10A). The d-band center of the $\text{Cu}/\text{C}_3\text{N}_4$ catalysts locates at -1.61 eV , which is significantly higher than that of Cu nanoparticles (-2.74 eV , Supplementary Figure 10B). Therefore, the CO adsorption energy is much lower on Cu single sites, leading to stronger bonding interactions between CO and Cu single sites (Refs: *J. Phys. Chem. C* **2009**, 113, 10548–10553, *ACS Catal.* **2023**, 13, 7822–7830). Furthermore, the unoccupied electron of $2\pi^*$ orbital of CO could partially shift under the Fermi levels and be occupied via backdonation of Cu 3d electron (Ref: *Phys. Rev. Lett.* **1996**, 76, 2141), leading to enhanced bond strength of Cu-CO (Supplementary Figure 10C). Therefore, there is a red shift of the CO vibration frequency on Cu single sites compared with that on the step/terrace sites.

For Figure 6 in Ref 21, the blue shift of the CO vibration frequency on Pt single atoms compared with the Pt clusters is also due to the different bonding interactions between CO molecules and Pt sites. The shift directions of CO vibration frequency are different on Cu and Pt surfaces when the dispersion of metals changes from clusters to single sites, which is likely due to the different electronic structures of Cu and Pt metals.

Action: We have attached the SEIRA spectra showing the CO adsorptions on C_3N_4 substrates in CO-saturated 0.5 M KHCO_3 as the Supplementary Figure 8 on Page 14 in the revised supplementary information. We have also added the following sentences on Page 6 in the revised manuscript when discussing the band at around 1890 cm^{-1} on the $\text{Cu}/\text{C}_3\text{N}_4$ catalysts and analyzing the bonding interactions between CO molecules and different Cu sites.

“SEIRA spectra collected on the C_3N_4 substrates show that there is no peak in the range from 1800 cm^{-1} to 2100 cm^{-1} , suggesting that there is no CO adsorption on the C or N sites of the C_3N_4 substrates (Supplementary Figure 8). Therefore, the band at around 1890 cm^{-1} is attributed to linearly bonded CO (CO_L) adsorbed on the Cu single sites of $\text{Cu}/\text{C}_3\text{N}_4$ catalysts (Figure 1, A and C). It is noted that the peak position of CO_L on Cu single sites is different from those on Cu step and terrace sites (Supplementary Figure 9). DFT calculations based on the Blyholder model show that the CO adsorption energy is lower on Cu single sites than that on metallic Cu sites (Supplementary Figure 10A), which leads to the red shift of the CO vibration frequency on Cu single sites compared with that on the step/terrace sites (Supplementary

Figure 10).²¹”.

We have also attached the CO adsorption energy as the Supplementary Figure 10 and added the following sentences on Page 16 in the revised supplementary information when analyzing the bonding interactions between CO molecules and different Cu sites.

“The d-band center of the Cu/C₃N₄ catalysts locates at -1.61 eV, which is significantly higher than that of Cu nanoparticles (-2.74 eV, Supplementary Figure 10B). Therefore, the CO adsorption energy is much lower on Cu single sites, leading to stronger bonding interactions between CO and Cu single sites.⁶ Furthermore, the unoccupied electron of 2π* orbital of CO could partially shift under the Fermi levels and be occupied via backdonation of Cu 3d electron,⁷ leading to enhanced bond strength of Cu-CO (Supplementary Figure 10C).”.

Supplementary Figure 8. Potential-dependent SEIRA spectra on the surfaces of C₃N₄ catalysts in the presence of CO-saturated 0.5 M KHCO₃ solution.

Figure R1. The CO adsorptions on the surfaces of Cu nanoparticles based on (A) Dewar-Chattock-Duncanson

model and (B) Blyholder model.

Figure R2. Simulated infrared spectra of CO adsorptions on (A) metallic Cu sites and (B) Cu single sites based on the Blyholder model.

Supplementary Figure 10. (A) CO adsorption energy for Cu/C₃N₄ SACs and Cu nanoparticles. (B) Calculated projected density of state (PDOS) of Cu 3d orbital. (C) Schematic illustration showing strong electron backdonation in the Cu/C₃N₄ SACs.

3) “The wavenumber of $\sim 2125\text{ cm}^{-1}$ is attributed to CO vibration interacting with K cationic species in electrical double layer; the further investigation is also needed to elucidate the origin of this band. Although the reference 26 is cited in here, in the reference 26, the control experiments of KHCO_3 produced by CO_2 saturating KOH on the Cu-poly surface are used to eliminate the role of electrolyte electrochemically deposited on the electrode, and confirming that this band corresponds to CO adsorbed on Cu sites.”

Response: We thank the reviewer for the valuable comments and we agree to the comments that further investigation is needed to elucidate the origin of this band at $\sim 2125\text{ cm}^{-1}$. Currently, the assignment of the band at $\sim 2125\text{ cm}^{-1}$ is still under debate in the scientific community. According to previous reports, the band at $\sim 2125\text{ cm}^{-1}$ is attributed to CO adsorptions either on reconstructed Cu sites or interacting with cationic species in the electrical double layer (e.g., K^+) (Refs: *J. Phys. Chem.* **1994**, *98*, 9577–9582, *J. Phys. Chem. B* **2006**, *110*, 22542–22550, *J. Phys. Chem. C* **2011**, *115*, 13312–13321). In the reference 26, Xu and co-workers conducted the control experiments on a gold film, as well as in KHCO_3 produced by CO_2 saturating KOH on the Cu-poly surfaces (Figure R3 in the response letter). Based on their results, there is no band at $\sim 2125\text{ cm}^{-1}$ in both cases. Therefore, the band at $\sim 2125\text{ cm}^{-1}$ is claimed to be corresponding to CO adsorptions on reconstructed Cu sites. However, if we take a closer look at Figure S4 in the reference 26 (Figure R3A in the response letter), there is still a tiny peak at $\sim 2125\text{ cm}^{-1}$, which might be ignored by the authors due to the baseline drift. To further confirm whether there is a peak at $\sim 2125\text{ cm}^{-1}$ on the surfaces of Au films, we have collected the SEIRA spectra in CO-saturated 0.05 M KOH and 0.5 M KHCO_3 using Au films (Supplementary Figure 14). It is found that the band at $\sim 2125\text{ cm}^{-1}$ appears on Au films in these two electrolytes. Considering that Au film was utilized as the substrate for the SEIRAS measurements, it is likely that the band at $\sim 2125\text{ cm}^{-1}$ is corresponding to CO vibration interacting with cationic species (K^+). We further collected the SEIRA spectra on the Cu-poly surfaces in CO-saturated 0.05 M KHCO_3 (Supplementary Figure 15A). There is no band at $\sim 2125\text{ cm}^{-1}$, which is consistent to the results in the reference 26. The absence of CO vibration interacting with cationic species (K^+) in 0.05 M KHCO_3 is likely due to the relatively low concentration of K^+ near the electrochemical interfaces. To confirm our hypothesis, SEIRAS measurements are conducted in CO-saturated 0.5 M KHCO_3 (Supplementary Figure 15B). It is noted that the band at $\sim 2125\text{ cm}^{-1}$ appears in 0.5 M KHCO_3 . Furthermore, SEIRAS results show that CO vibration interacting with cationic species appears in CO-saturated 0.05 M KOH (Supplementary Figure 15C), which is consistent to the results in the reference 26. Despite the relatively low bulk concentration of K^+ in 0.05 M KOH, the local concentration of K^+ near the electrochemical interface could be high enough to show the CO vibration in SEIRA spectra due to the relatively low absolute potential. Therefore, we believe that the band at $\sim 2125\text{ cm}^{-1}$ is corresponding to CO vibration interacting with cationic species (K^+), which is also consistent to previous reports (Refs: *J. Phys. Chem. B* **2006**, *110*, 22542–22550, *J. Phys. Chem. C* **2011**, *115*, 13312–13321).

Action: We have added the following sentences and also updated the cited references when discussing the band at $\sim 2125\text{ cm}^{-1}$ on Page 7 in the revised manuscript.

“It is noted that the high wavenumber component (gray, $\sim 2125\text{ cm}^{-1}$) appears after CO_2RR at -1.2 V for 20 min. Currently, the assignment of the high wavenumber component is still under debate.^{26,29–31} To elucidate the origin of the high wavenumber component, control experiments are conducted with a gold film in CO-saturated 0.05 M KOH and 0.5 M KHCO_3 (Supplementary Figure 14). The band at $\sim 2125\text{ cm}^{-1}$ appear on the Au film, suggesting that it is corresponding to the CO vibration interacting with K cationic species in the electrical double layer. Further investigations with Cu films in CO-saturated 0.05 M KHCO_3 , 0.5 M KHCO_3 , and 0.05 M KOH confirm that this band corresponds to CO vibration interacting with K cationic species (Supplementary Figure 15).^{30,31}”

We have also attached the SEIRA spectra on the surfaces of Au and Cu films in different electrolytes as

Supplementary Figure 14 and 15 on Page 20 and 21 in the revised supplementary information, respectively.

Figure R3. (A) Operando ATR-SEIRA spectra of a gold film in CO-saturated 0.05 M KOH in Figure S4 in the reference 26. (B) Operando ATR-SEIRA spectra of a Cu film in CO-saturated 0.05 M KOH in Figure S5 in the reference 26 (Ref: *ACS Catal.* **2019**, *9*, 474–478).

Supplementary Figure 14. Time-dependent SEIRA spectra on the surfaces of Au films at -0.4 V vs RHE in (A) CO-saturated 0.05 M KOH solution and (B) CO-saturated 0.5 M KHCO₃ solution.

Supplementary Figure 15. Time-dependent SEIRA spectra on the surfaces of Cu films at -0.4 V vs RHE in (A) CO-saturated 0.05 M KHCO_3 solution, (B) CO-saturated 0.5 M KHCO_3 solution, and (C) CO-saturated 0.05 M KOH solution.

4) “Line 167-169, the Cu(200) and Cu(111) facets are verified by lattice distances, and two facets used to explain the CO adsorptions on Cu step and terrace sites, respectively. while, the single crystal surface of Cu(200) is same as the Cu(100), both Cu(100) and Cu(111) are regarded as the low-index surfaces, as the reference 29 said, the Cu(211), Cu(221), and Cu(532) are the regular step and kink surfaces.”

Response: We thank the reviewer for the valuable comments. Indeed, the single crystal surface of Cu(200) is same as the Cu(100). Typically, the low-index surfaces in face-centered cubic Cu nanocrystals include Cu(100), Cu(110), and Cu(111) facets. According to previous reports including the reference 35 in the revised manuscript (Refs: *J. Electrochem. Soc.* **1997**, *144*, L261, *J. Phys. Chem. C* **2014**, *118*, 26103–26114, *J. Phys. Chem. C* **2015**, *119*, 251–261, *J. Phys. Chem. B* **2018**, *122*, 963–971), there are both terrace and step edge sites in the low-index facets. The high-index facets such as Cu(211), Cu(221), and Cu(532) can be resolved into the sum of several low-index facets (Refs: *Phys. Rev. B* **2005**, *71*, 035402, *J. Phys. Chem. C* **2008**, *112*, 11086–11089). Therefore, there are more step and kink sites in high-index facets as compared with low-index facets as reported in the reference 29.

Action: We have updated the cited references when discussing the lattice distances of Cu(200) and Cu(111) facets on Page 8 in the revised manuscript.

“The lattice distances of 0.18 and 0.21 nm are corresponding to the Cu(200) and Cu(111) facets,^{32,33} which are in good agreements with the CO adsorptions on Cu step and terrace sites, respectively.^{34–37”}

5) “The geometrical structure of CO adsorption on metallic and monoatomic Cu sites Cu/C₃N₄, CuPc, Cu-NC, and Cu-SNC is necessary.”

Response: We thank the reviewer for the constructive comments. We have conducted DFT calculations to

study the geometrical structure of CO adsorption on metallic and monoatomic Cu sites in the Cu/C₃N₄, CuPc, Cu-NC, and Cu-SNC catalysts (Supplementary Figure 50 and 51). The simulated infrared spectra of CO adsorption on metallic and monoatomic Cu sites are in good agreement with the SEIRAS measurements and previous reports (Refs: *Sci. Adv.* **2020**, *6*, eabd2569, *Nat. Commun.* **2023**, *14*, 6164).

Action: We have attached the geometrical structure of CO adsorption on metallic and monoatomic Cu sites, as well as the simulated infrared spectra, as Supplementary Figure 50 and 51 on Page 57 and 58 in the revised supplementary information. Also, we have added the following sentences when discussing the CO adsorption on metallic and monoatomic Cu sites on Page 12 in the revised manuscript.

“Simulated infrared spectra of CO adsorption show that obvious peaks corresponding to CO_L on metallic Cu sites and Cu single sites are located at around 2063 cm⁻¹ and 1964 cm⁻¹, respectively, which are consistent to the SEIRAS measurements and previous reports (Supplementary Figure 50 and 51).^{36,41}”.

Supplementary Figure 50. Simulated infrared spectra of CO adsorption on different metallic Cu surfaces. (A) C₃N₄ supported Cu nanoparticles. (B) Cu(100) facet. (C) Cu(110) facet. (D) Cu(111) facet. The inset shows the corresponding geometric configuration of CO adsorption.

Supplementary Figure 51. Simulated infrared spectra of CO adsorption on different Cu single sites. (A) Cu/C₃N₄ SACs. (B) CuPc SACs. (C) Cu-NC SACs. (D) Cu-SNC SACs. The inset shows the corresponding geometric configuration of CO adsorption.

6) “*Ex-situ X-ray diffraction patterns demonstrate that small portion of Cu single site are converted into metallic Cu sites after CO₂RR for 8h, but the C₂₊ products start to form after the CO₂RR for 30 min on the Cu/C₃N₄.*”

Response: We thank the reviewer for the valuable comments. The reactivity measurements show that C₂₊ products including C₂H₄ and C₂H₅OH start to form after the CO₂RR for 30 min on the Cu/C₃N₄ SACs. The original spectra of C₂₊ product peaks in NMR and GC on the surfaces of Cu/C₃N₄ catalysts are included in Supplementary Figure 39 and 43 on Page 46 and 50 in the revised supplementary information, respectively. There are no detectable C₂₊ products after the CO₂RR for 15 min and decent amount of C₂₊ products are detected after the CO₂RR for 30 min. The reactivity results are in good agreement with the SEIRA spectra that the structural evolution starts after the CO₂RR for 20 min. HAADF-STEM images of the Cu/C₃N₄ SACs post reaction show that there are Cu nanoparticles formed after CO₂RR at -1.2 V for 8 h (Figure 3A), which is consistent to the SEIRAS results of CO adsorptions on metallic Cu sites. According to previous reports (Refs: *ACS Catal.* **2018**, *8*, 7809–7819, *Ind. Eng. Chem. Res.* **2019**, *58*, 19434–19445), the detection

limit of X-ray diffraction technique is around 5 nm. If there is significant conversion of Cu single sites into Cu nanoparticles with a size larger than 5 nm, diffraction peaks corresponding to Cu nanoparticles would show up in the X-ray diffraction patterns. The bulk crystal structure of Cu/C₃N₄ catalysts remains the same after the CO₂RR for 8 h (Supplementary Figure 12), demonstrating that there are no significant aggregations of Cu single sites into Cu nanoparticles.

Action: We have attached the original spectra of C₂₊ product peaks in NMR and GC on the surfaces of Cu/C₃N₄ catalysts as Supplementary Figure 39 and 43 on Page 46 and 50 in the revised supplementary information and also added the following sentences when discussing the *ex-situ* X-ray diffraction patterns on Page 6 in the revised manuscript.

“According to previous reports,^{27,28} the detection limit of X-ray diffraction technique is around 5 nm. If there is significant conversion of Cu single sites into large Cu nanoparticles, diffraction peaks corresponding to Cu nanoparticles would show up in the X-ray diffraction patterns. *Ex-situ* X-ray diffraction patterns show that the bulk crystal structure of Cu/C₃N₄ catalysts remains the same after being sprayed onto carbon paper and after the CO₂RR for 8 h (Supplementary Figure 12), demonstrating that there are no significant aggregations of Cu single sites into large Cu nanoparticles.”

Supplementary Figure 39. Typical ¹H NMR spectra of C₂₊ products with Cu/C₃N₄ SACs as the catalysts at -1.2 V vs RHE for (A–C) 30 min, (D) 15 min. A, B, and C show the ¹H NMR spectra in three parallel experiments.

Supplementary Figure 43. Typical GC traces of C_2+ products with Cu/C_3N_4 SACs as the catalysts at -1.2 V vs RHE for (A–C) 30 min, (D) 15 min. A, B, and C show the GC traces in three parallel experiments.

7) “Although the reconstruction process of Cu SACs convert into Cu nanoparticles is found by using the ATR-SEIRAS, a physical image of adsorbed CO promoting the Cu aggregation is not clear.”

Response: We thank the reviewer for the valuable comments. DFT calculations are conducted to show the reconstruction process of Cu SACs into Cu nanoparticles (Supplementary Figure 53 and 54). It is revealed that the adsorption of H is a vital driving force for the leaching of Cu single sites from the catalyst surfaces. The adsorption of H on the Cu SACs becomes stronger with the decreasing potentials, leading to the leaching of Cu single sites by weakening the $Cu-N$ bonds. The collision of the Cu atoms forms a transient Cu cluster, which is consistent to previous reports (Ref: *J. Am. Chem. Soc.* **2022**, *144*, 17140–17148).

Action: We have added the following sentences when discussing reconstruction of Cu SACs with DFT calculations on Page 12 in the revised manuscript.

“The adsorption of H is revealed to be a vital driving force for the leaching of Cu single sites from the catalyst surfaces (Supplementary Figure 53). The adsorption of H on the Cu SACs becomes stronger with the decreasing potentials, leading to the leaching of Cu single sites by weakening the $Cu-N$ bonds. The collision of the Cu atoms forms a transient Cu cluster (Supplementary Figure 54), which is consistent to previous reports.⁴³”.

Supplementary Figure 53. The configuration of (A) *H on N sites, (B) *CO on Cu single site, and (C) *H on Cu single site. (D) The corresponding free energy for the three adsorptions. The adsorption energy of *H on N site is the lowest, indicating that the *H on the N site is most likely to promote the breaking of Cu-N bonds.

Supplementary Figure 54. Free energy of leaching a Cu atom from Cu/C₃N₄ SACs. The transient Cu nanoparticle is unstable which would further induce the breaking of Cu-N bonds. Due to the high computational cost, the applied potential is not considered during the calculations.

8) “The TOC is very confusing of the structure of two adjacent CO on Cu nanoparticle.”

Response: We thank the reviewer for the valuable comments and have revised the TOC accordingly. Two adjacent CO molecules on Cu nanoparticles are bonded with two adjacent Cu atoms, which is consistent to previous reports (Ref: *Sur. Sci.* **2016**, 654, 56–62).

Action: We have updated the TOC on Page 21 in the revised manuscript to make sure it is consistent to previous reports.

Reviewer #2

Recommendation:

“The authors conducted a comprehensive investigation on the structural evolution of Cu single-atom catalysts (SACs) during the electrochemical CO₂ reduction. They employed operando infrared spectroscopic measurements to monitor CO adsorption on both single atom and metal cluster sites. According to their findings, the Cu SACs initially generated C₁ product (CO) owing to the absence of C-C coupling. However, as the reaction progressed, they observed the gradual formation of C₂ products due to the segregation of Cu SACs into metallic nanoparticles. Another important observation was that the stability of SACs proved to be significantly influenced by the substrates, attributed to differences in coordination. This work is well organized and the interpretation of experimental results is executed well. Nevertheless, to ensure acceptance, a major revision is recommended to address certain aspects.”

Response:

We thank the reviewer for his/her positive assessments and have revised our manuscript accordingly.

1) “Elaborate on the segregation process of single-atom catalysts (SACs): Provide a more comprehensive explanation of the process. What drives the breaking of Cu-N coordination bonds and the formation of Cu-Cu metallic bonds?”

Response: We thank the reviewer for the valuable comments. DFT calculations are conducted to show the reconstruction process of Cu SACs into Cu nanoparticles (Supplementary Figure 53 and 54). It is revealed that the adsorption of H is a vital driving force for the leaching of Cu single sites from the catalyst surfaces. The adsorption of H on the Cu SACs becomes stronger with the decreasing potentials, leading to the leaching of Cu single sites by weakening the Cu-N bonds. The collision of the Cu atoms forms a transient Cu cluster, which is consistent to previous reports (Ref: *J. Am. Chem. Soc.* **2022**, *144*, 17140–17148).

Action: We have added the following sentences when discussing reconstruction of Cu SACs with DFT calculations on Page 12 in the revised manuscript.

“The adsorption of H is revealed to be a vital driving force for the leaching of Cu single sites from the catalyst surfaces (Supplementary Figure 53). The adsorption of H on the Cu SACs becomes stronger with the decreasing potentials, leading to the leaching of Cu single sites by weakening the Cu-N bonds. The collision of the Cu atoms forms a transient Cu cluster (Supplementary Figure 54), which is consistent to previous reports.⁴³”

Supplementary Figure 53. The configuration of (A) *H on N sites, (B) *CO on Cu single site, and (C) *H on Cu single site. (D) The corresponding free energy for the three adsorptions. The adsorption energy of *H on N site is the lowest, indicating that the *H on the N site is most likely to promote the breaking of Cu-N bonds.

Supplementary Figure 54. Free energy of leaching a Cu atom from Cu/C₃N₄ SACs. The transient Cu nanoparticle is instable which would further induce the breaking of Cu-N bonds. Due to the high computational cost, the applied potential is not considered during the calculations.

2) “Not only single-atom catalysts (SACs) but also Cu nanoparticle catalysts tend to aggregate during electrochemical reactions due to various factors such as reaction intermediate-metal bonding [Science **351**, 475 (2016)], potential [Nat. Commun. **9**, 1 (2018)], gas evolution [Nat. Energy **7**, 537 (2022)], or cathodic corrosion [ACS Catal. **12**, 13174 (2022)]. This broad aspect of catalyst aggregation can significantly impact catalytic performance and stability. To address these challenges, it is highly recommended to discuss potential strategies that can effectively mitigate catalyst aggregation, taking into account the insights gained from your results and also citing the relevant references.”

Response: We thank the reviewer for the valuable comments and fully agree to the reviewer’s opinion. Not only single-atom catalysts (SACs) but also Cu nanoparticle catalysts tend to aggregate during electrochemical reactions due to various factors such as reaction intermediate-metal bonding, potential, gas evolution, and cathodic corrosion (Ref: Science **2016**, 351, 475, Nat. Commun. **2018**, 9, 3117, Nat. Energy **2022**, 7, 537, ACS Catal. **2022**, 12, 13174), which can significantly impact catalytic performance and stability. In the past few year, different strategies have been developed to achieve improved catalytic performance with high stability, including the synthesis of electrocatalysts with different shapes, compositions and structures, coating the electrocatalysts with ultrathin carbon shells, and etc. (Ref: Nano Lett. **2016**, 16, 6644–6649, J. Am. Chem. Soc. **2015**, 137, 15478–15485) We are more than happy to discuss these strategies that can effectively mitigate catalyst aggregation and improve the catalyst stability, which has been added on Page 12 in the revised manuscript.

Action: We have added the following sentences when discussing potential strategies that can effectively mitigate catalyst aggregation on Page 12 in the revised manuscript.

“The adsorption of H is revealed to be a vital driving force for the leaching of Cu single sites from the catalyst surfaces (Supplementary Figure 53). The adsorption of H on the Cu SACs becomes stronger with the decreasing potentials, leading to the leaching of Cu single sites by weakening the Cu-N bonds. The collision of the Cu atoms forms a transient Cu cluster (Supplementary Figure 54), which is consistent to previous reports.⁴³ It is also noted that not only SACs but also nanoparticle catalysts tend to aggregate during electrochemical reactions due to various factors such as reaction intermediate-metal bonding, potential, gas evolution, and cathodic corrosion,^{44–47} which can significantly impact catalytic performance and stability. In the past few year, different strategies have been developed to achieve improved catalytic performance with high stability, including the synthesis of electrocatalysts with different shapes, compositions and structures, coating the electrocatalysts with ultrathin carbon shells, and *etc.*^{48,49”}.

3) *“The faradaic efficiency of C₂₊ products is observed to be lower than 1% (Figure S31), and the production rate is also found to be lower than 0.04 umol/min (Figure S32). These values appear to be very small and may raise concerns about reliability. Additionally, it is amazing that the observed trends of catalytic activity, with Cu/C₃N₄ > CuPc > Cu-NC > Cu-SNC, are quite remarkable, especially given the very small error range (FEs < 0.2%). To ensure the reliability of these results, please provide further evidence, such as the original spectrum of C₂ product peaks in NMR or gas chromatograph data.”*

Response: We thank the reviewer for the valuable comments. We have attached the original ¹H NMR spectra and GC profiles of C₂₊ products (Supplementary Figure 39–46), as well as the chronoamperometry profiles (Supplementary Figure 47), on Page 46–54 in the revised supplementary information. The ¹H NMR, GC, and electrochemical results show good reproducibility with very small error range, which confirms that the reactivity results in our work is reliable.

Action: We have attached the original ¹H NMR spectra and GC profiles of C₂₊ products (Supplementary Figure 39–46), as well as the chronoamperometry profiles (Supplementary Figure 47), on Page 46–54 in the revised supplementary information.

Supplementary Figure 39. Typical ^1H NMR spectra of C_2^+ products with $\text{Cu}/\text{C}_3\text{N}_4$ SACs as the catalysts at -1.2 V vs RHE for (A–C) 30 min, (D) 15 min. A, B, and C show the ^1H NMR spectra in three parallel experiments.

Supplementary Figure 40. Typical ^1H NMR spectra of C_2^+ products with CuPc SACs as the catalysts at -1.2 V vs RHE for (A–C) 60 min, (D) 40 min. A, B, and C show the ^1H NMR spectra in three parallel experiments.

Supplementary Figure 41. Typical ^1H NMR spectra of C_{2+} products with Cu-NC SACs as the catalysts at -1.2 V vs RHE for (A–C) 200 min, (D) 150 min. A, B, and C show the ^1H NMR spectra in three parallel experiments.

Supplementary Figure 42. Typical ^1H NMR spectra of C_2^+ products with Cu-SNC SACs as the catalysts at -1.2 V vs RHE for (A–C) 300 min, (D) 240 min. A, B, and C show the ^1H NMR spectra in three parallel experiments.

Supplementary Figure 43. Typical GC traces of C_2^+ products with Cu/ C_3N_4 SACs as the catalysts at -1.2 V vs RHE for (A–C) 30 min, (D) 15 min. A, B, and C show the GC traces in three parallel experiments.

Supplementary Figure 44. Typical GC traces of C_2+ products with CuPc SACs as the catalysts at -1.2 V vs RHE for (A–C) 60 min, (D) 40 min. A, B, and C show the GC traces in three parallel experiments.

Supplementary Figure 45. Typical GC traces of C_2+ products with Cu-NC SACs as the catalysts at -1.2 V vs RHE for (A–C) 200 min, (D) 150 min. A, B, and C show the GC traces in three parallel experiments.

Supplementary Figure 46. Typical GC traces of C_2+ products with Cu-SNC SACs as the catalysts at -1.2 V vs RHE for (A–C) 300 min, (D) 240 min. A, B, and C show the GC traces in three parallel experiments.

Supplementary Figure 47. The chronoamperometry profiles of different Cu SACs in 0.5 M KHCO_3 electrolyte at an applied potential of -1.2 V vs RHE. (A) Cu/ C_3N_4 for 30 min. (B) CuPc for 60 min. (C) Cu-NC for 200 min. (D) Cu-SNC for 300 min. The green, blue, and purple curves show the chronoamperometry profiles in three parallel experiments.

Reviewer #3

Recommendation:

“The manuscript by Zhang et al. reported the quantitatively monitoring of the structure evolution of Cu single-atom catalysts during the electrochemical reduction of CO_2 . The evolution rate of Cu single-atom catalysts on different substrates were quantified using surface-enhanced infrared absorption spectroscopy (SEIRAS). Combined with DFT calculations, it was found that the stability of Cu SACs is highly dependent on their formation energy, which can be manipulated by controlling the affinity between Cu sites and substrates. I enjoy very much on reading this manuscript, and the method used by the authors might be highly useful for others to study the catalyst stability of single atom catalysts. I would like to recommend the publication of this manuscript after some minor concerns are fully addressed.”

Response:

We thank the reviewer for his/her positive assessments and have revised our manuscript accordingly.

1) “As far as I know, it is challenging to use ATR SEIRAS to quantitatively measure the chemical kinetics of the reactions of adsorbates on catalyst surface, especially with various particle size and morphology. I suggest the authors claim the limitations of their method for the calculation of chemical kinetics.”

Response: We thank the reviewer for the constructive comments. Indeed, it is challenging to use ATR-SEIRAS to quantitatively measure the reaction kinetics on the catalyst surfaces, which is primarily due to the lacking of control on the size and morphology of metal films. In this work, the amount of Cu single sites is relatively stable during the CO₂RR. Therefore, the CO_L on Cu single sites is utilized as the internal standard to eliminate the variations caused by the poor reproducibility of metal films. We have achieved the quantification of CO adsorptions on metallic Cu sites using the internal standard method (Figure 2). To further broaden the applications of ATR-SEIRAS in quantitative measurements, we can introduce another IR active compound as the internal standard or control the size/morphology of metal films. This is something that we are working on right now and we are more than happy to share some preliminary results here. We have fabricated highly ordered rhombic gold nanocube superlattices (GNSs) as substrates for ATR-SEIRAS with improved sensitivity and reproducibility (Figure R4 and R5 in the response letter). The GNSs could be potentially utilized for further quantitative measurements.

Action: We have added the following sentences when discussing the limitations of ATR-SEIRAS for the calculation of chemical kinetics on Page 13 in the revised manuscript.

“However, the relatively poor sensitivity and reproducibility of chemically deposited metal films make it challenging to achieve quantitative measurements of reaction kinetics using ATR-SEIRAS technique. We believe that the fabrication of SEIRAS substrates with uniform size and well-defined morphology would be a promising direction for achieving quantitative understanding of reaction mechanisms.”.

Figure R4. The SEM image of highly ordered rhombic gold nanocube superlattices (GNSs) on Si prism for ATR-SEIRAS.

Figure R5. Potential-dependent SEIRA spectra of adsorbed CO bands on three different GNSs in CO-saturated 0.1 M HClO₄: **a** GNS-1, **b** GNS-2, and **c** GNS-3. Cyclic voltammograms showing the ECSAs of GNSs: **d** GNS-1, **e** GNS-2, and **f** GNS-3.

2) “In terms of the kinetic analysis, the authors may want to consider the aggregation of reduced Cu(0) for the formation of Cu nanoparticles. Also, the authors need to clarify that the density of adsorbed CO molecules on the surfaces of Cu nanoparticles is a constant based on their analysis.”

Response: We thank the reviewer for the valuable comments. Indeed, there are two steps in the aggregation process of Cu SACs for the formation of Cu nanoparticles: 1) reduction of Cu single sites into zero-valent Cu atoms due to the breaking of Cu-N bonds, 2) aggregation of zero-valent Cu atoms for the formation of Cu nanoparticles. It has been proved that the Cu-N breaking step is the rate-limiting step in the reconstruction process of Cu SACs into Cu nanoparticles, which is also consistent to previous reports on the synthesis of metal nanoparticles (Refs: *Angew. Chem. Int. Ed.* **2008**, *48*, 60–103, *Proc. Natl. Acad. Sci. U.S.A.* **2017**, *114*, 13619–13624). Due to the high surface energy of Cu(0), the reaction kinetics of the aggregation process for the formation of Cu nanoparticles is quite fast. Therefore, it is reasonable to simplify the reconstruction process for the kinetic analysis by only considering Cu-N breaking step.

When we determine the reaction kinetics of the reconstruction process, the density of adsorbed CO molecules on the surfaces of 2-nm Cu nanoparticles was reasonably set as a constant. We have clarified it on Page 23 in the revised supplementary information.

Action: We have added the following sentences when discussing the procedure for the quantification of evolution rate on Page 23 in the revised supplementary information.

“The density of adsorbed CO molecules on the surfaces of 2-nm Cu nanoparticles is set as a constant.”.

3) “For the quantification, will the formed Cu nanoparticles affect the signal of SEIRA spectra, since chemically deposited Cu films consisting of Cu nanoparticles are widely utilized as the substrates for SEIRAS studies?”

Response: We thank the reviewer for his/her constructive comments. Chemically deposited Cu films made of Cu nanoparticles have been widely used as the substrates for SEIRAS because of their SEIRA effect. The size of Cu nanoparticles in these Cu films is around 50–150 nm according to previous reports (Refs: *Chem. Lett.* **2004**, 33, 278–279, *Electrochim. Acta* **2007**, 52, 5950–5957, *ACS Cent. Sci.* **2016**, 2, 522–528). In contrast, the average size of formed Cu nanoparticles is around 2 nm in this work. Control experiments are conducted to check whether the formed 2-nm Cu nanoparticles would affect the signal of SEIRA spectra. The FTIR results of commercial Cu catalysts with an average size of several hundreds of nanometers show that there is no IR absorption in the range from 2000 cm^{-1} to 2200 cm^{-1} , indicating that the SEIRA effect of Cu films is primarily due to the local electromagnetic field (Figure R6 in the response letter). SEIRA spectra collected on thinner Cu films with smaller particle sizes show that there are no CO adsorption peaks in the potential range from –0.1 to –0.9 V vs RHE, suggesting that the formed 2-nm Cu nanoparticles would not affect the signal of SEIRA spectra (Figure R7 in the response letter). The finite difference time domain (FDTD) simulations show that the electromagnetic field strength near the 2-nm Cu nanoparticles is even weaker than that of the microenvironment, indicating that the SEIRA effect of the 2-nm Cu nanoparticle is negligible (Supplementary Figure 20).

Action: We have added the following sentences when discussing the formed 2-nm Cu nanoparticles on Page 8 in the revised manuscript and also included the FDTD simulation results as Supplementary Figure 20 on Page 27 in the revised supplementary information.

“High-resolution transmission electron microscopy (HRTEM) image reveals that the size of as-formed Cu nanoparticles is around 2 nm (Figure 3B), which shows negligible effect on the SEIRAS according to the FDTD simulations (Supplementary Figure 20).”.

Figure R6. Fourier transform infrared (FTIR) spectra of commercial Cu catalysts.

Figure R7. The SEM images of Cu films with different thicknesses and the corresponding SEIRA spectra showing the CO adsorption in the range of 1800–2200 cm^{-1} . (A and B) SEM images of thin Cu films. (C and D) SEM images of regular Cu films. (E) The SEIRA spectra showing the CO adsorption in the range of 1800–2200 cm^{-1} on Cu films with different thicknesses (Black solid line refers to regular Cu films and dash line refers to the thin Cu films).

Supplementary Figure 20. The finite difference time domain (FDTD) simulations of the local electromagnetic field for 2-nm Cu nanoparticles. (A) 3D model of 2-nm Cu nanoparticles. Simulated near-field enhancement $|E|^2$ of for 2-nm Cu nanoparticles on (B) xy plane, (C) xz plane and (D) yz plane.

4) *“I would like to suggest the authors to add the proposed structures of all four copper catalysts in the main text, especially for the Cu/SNC catalyst.”*

Response: We thank the reviewer for his/her valuable comments and have added the structures of all four copper catalysts in Figure 4C on Page 11 in the revised manuscript.

Action: We have added the structures of all four copper catalysts in Figure 4C on Page 11 in the revised manuscript.

Figure 4. Structure-stability relations of Cu SACs. (A) Potential-dependent evolution rates of Cu single sites to metallic Cu sites on Cu/C₃N₄, CuPc, Cu-NC, and Cu-SNC SACs. (B) Potential-dependent production rates of C₂H₄ and C₂H₅OH on Cu/C₃N₄, CuPc, Cu-NC, and Cu-SNC SACs. (C) DFT calculations of CO adsorption energy and formation energy for Cu/C₃N₄, CuPc, Cu-NC, and Cu-SNC SACs.

5) “For the proposed chemical structure of Cu/C₃N₄, did the authors consider Cu-N₂-OH coordination configuration where Cu is coordinating with the edge NN sites and an OH group? To me, the currently proposed structure of Cu-N₃ is with high structural reorganization, and the coordinating N sites is out of the C₃N₄ plan. Considering the low Cu loading and poor stability of the catalyst, edge NN sites should not be ruled out. DFT calculations should be carried out to address this issue.”

Response: We thank the reviewer for his/her valuable comments and have conducted DFT calculations to further investigate the possibility of Cu-N₂-OH coordination configuration, as well as the Cu-N₂C, and Cu-N₄ configurations (Supplementary Figure 6 on Page 12 in the revised supplementary information). It is noted that the calculated XANES spectra of the optimized Cu-N₃ structure are in good agreement with the experimental results (Supplementary Figure 4A and Figure 6 in the revised supplementary information). In contrast, there is a large discrepancy in terms of peak position and intensity between the calculated XANES spectra of other optimized geometrical structures (Cu-N₂OH, Cu-N₂C, and Cu-N₄) and experimental results (Supplementary Figure 4A and Figure 6 in the revised supplementary information). Therefore, we believe that the optimized Cu-N₃ structure in the revised manuscript could represent the geometrical structure of the Cu/C₃N₄ catalysts.

Action: We have added the calculated XANES spectra as Supplementary Figure 6 on Page 12 in the revised supplementary information and added the following sentences on Page 5 in the revised manuscript when discussing the optimization of Cu/C₃N₄ structures:

“Four possible geometrical structures of Cu/C₃N₄ catalysts including Cu-N₃, Cu-N₂OH, Cu-N₂C, and Cu-N₄ are investigated using DFT calculations (Supplementary Figure 6). It is noted that the calculated XANES spectra of the optimized Cu-N₃ structure are in good agreement with the experimental results (Supplementary Figure 4A and Figure 6). In contrast, there is a large discrepancy in terms of peak position and intensity between the calculated XANES spectra of other optimized geometrical structures and experimental results (Supplementary Figure 4A and Figure 6). Therefore, Cu-N₃ coordination is likely the geometrical structure of the Cu/C₃N₄ catalysts.”

Supplementary Figure 6. Simulated XANES spectra for four possible geometrical structures of Cu/C₃N₄ catalysts. (A) Cu-N₃. (B) Cu-N₂OH. (C) Cu-N₂C. (D) Cu-N₄.

6) “Line 132. Read “*Ex-situ X-ray diffraction patterns show that the bulk crystal structure of Cu/C₃N₄ catalysts remains the same after being sprayed onto carbon paper and after the CO₂RR for 8 h (Supplementary Figure 9), demonstrating that a small portion of Cu single sites are converted into metallic Cu sites.*” Amorphous sub-nano Cu clusters could not be ruled out by X-ray diffraction and thereby it is not safe to mention only a small portion of Cu single sites are converted into metallic sites. Especially, Figure 3A displays more metallic copper atoms than single Cu atoms.”

Response: We appreciate the reviewer for his/her constructive comments. We agree to the reviewer’s opinion that amorphous sub-nano Cu clusters could not be ruled out by X-ray diffraction technique. *Ex-situ* X-ray diffraction patterns were utilized to exclude the existence of Cu nanoparticles with larger sizes. The HAADF-STEM image in Figure 3A only shows a small portion of Cu SACs. We have included more TEM images as Supplementary Figure 19 and 33 on Page 26 and 40 in the revised supplementary information, which could represent a more complete picture of the Cu SACs post reaction. There is no severe conversion of Cu SACs into Cu nanoparticles based on the TEM analysis.

Action: We have made a safer statement when discussing the *ex-situ* XRD results on Page 6 in the revised manuscript and also included more TEM images as Supplementary Figure 19 and 33 on Page 26 and 40 in the revised supplementary information.

“*Ex-situ* X-ray diffraction patterns show that the bulk crystal structure of Cu/C₃N₄ catalysts remains the same after being sprayed onto carbon paper and after the CO₂RR for 8 h (Supplementary Figure 12), demonstrating that there are no significant aggregations of Cu single sites into large Cu nanoparticles.”

Supplementary Figure 19. HAADF-STEM images of the Cu/C₃N₄ SACs post CO₂RR at -1.2 V for 8 h. There is decent amount of Cu single sites after the CO₂RR for 8 h, indicating that the evolution of Cu single atoms to Cu nanoparticles is minor.

Supplementary Figure 33. HAADF-STEM images of (A–C) CuPc, (D–F) Cu-NC and (G–I) Cu-SNC SACs post CO₂RR at –1.2 V for 8 h (selected single Cu atoms and nanoparticles are marked by the orange circles and white ellipses, respectively).

REVIEWERS' COMMENTS

Reviewer #1 (Remarks to the Author):

Thanks the authors for taking into consideration of all my comments, I have carefully checked the revised manuscript and the author's response. The modifications are sufficient and reasonable. I can now recommend it for publication on Nature Commination.

Reviewer #2 (Remarks to the Author):

The authors have responded to all the queries.

It is ready for the publication.

Reviewer #3 (Remarks to the Author):

The authors have fully addressed my concerns. I have no more comments on this manuscript.